



# Improving Ensemble Data Assimilation through Probit-space Ensemble Size Expansion for Gaussian Copulas (PESE-GC)

Man-Yau Chan[1,2]

[1]Department of Geography, The Ohio State University, Columbus, Ohio, 43210, USA
[2]Advanced Study Program, National Center for Atmospheric Research, Boulder, Colorado, USA

**Correspondence:** Man-Yau Chan (chan.1063@osu.edu)

**Abstract.** Small forecast ensemble sizes ($< 100$) are common in the ensemble data assimilation (EnsDA) component of geophysical forecast systems, thus limiting the error-constraining power of EnsDA. This study proposes an efficient and embarrassingly parallel method to generate additional ensemble members: the Probit-space Ensemble Size Expansion for Gaussian Copulas (PESE-GC; "peace gee see"). Such members are called "virtual members". PESE-GC utilizes the users' knowledge of the marginal distributions of forecast model variables. Virtual members can be generated from any (potentially non-Gaussian) multivariate forecast distribution that has a Gaussian copula. PESE-GC's impact on EnsDA is evaluated using the 40-variable Lorenz 1996 model, several EnsDA algorithms, several observation operators, a range of EnsDA cycling intervals and a range of forecast ensemble sizes. Significant improvements to EnsDA ($p < 0.01$) are observed when either 1) the forecast ensemble size is small ($\leq 20$ members), 2) the user selects marginal distributions that improves the forecast model variable statistics, and/or 3) the rank histogram filter is used with non-parametric priors in high forecast spread situations. These results motivate development and testing of PESE-GC for EnsDA with high-order geophysical models.

## 1 Introduction

Geophysical forecast models are often computationally expensive to run. As a result, geophysical ensemble data assimilation (EnsDA) typically uses $<100$ forecast ensemble members. Such small forecast ensemble sizes result in two EnsDA-degrading effects: 1) sampling errors contaminate the ensemble-estimated flow-dependent statistics, and, 2) for certain EnsDA methods, small forecast ensembles result in limited representation of observation likelihood functions. As such, low-cost methods that introduce additional ensemble members (henceforth, "virtual members") to the original forecast members (henceforth, "forecast members") have the potential to improve EnsDA.

Several types of ensemble expansion methods have been proposed in the literature, all of which have strengths and weaknesses. The first type is random draws from climatology (Castruccio et al., 2020; El Gharamti, 2020; Lei et al., 2021). Though computationally efficient, this type of ensemble expansion method cannot generate members with flow-dependent ensemble statistics.

An alternative type of ensemble expansion method is to aggregate forecast ensemble members across time (Xu et al., 2008; Yuan et al., 2009; Huang and Wang, 2018; Gasperoni et al., 2022). Though this type of method efficiently produces members



with flow-dependent statistics, the number of virtual members created is limited (Huang and Wang, 2018; Gasperoni et al., 2022).

A third type of ensemble expansion method is to search a historical catalog for forecast states similar to the current forecast (Van den Dool, 1994; Tippett and Delsole, 2013; Monache et al., 2013; Wan and Van Der Merwe, 2000; Grooms, 2021). The virtual members resulting from this search have flow-dependent statistics. However, such methods are typically expensive to

employ (e.g., Grooms (2021)).

Ensemble modulation (Bishop and Hodyss, 2009, 2011; Bishop et al., 2017; Kotsuki and Bishop, 2022; Wang et al., 2021) is the fourth type of ensemble expansion method. Such methods expand ensembles by combining a localization matrix with the original ensembles' perturbations (see Bishop and Hodyss (2009) for details). However, ensemble modulation assumes that forecast uncertainties have Gaussian statistics. This limits the usefulness of ensemble modulation to Gaussian situations.

Furthermore, even if the forecast ensemble has Gaussian statistics, the expanded ensemble can have non-Gaussian statistics (see the supplement). This non-Gaussianity contradicts ensemble modulation's assumption of Gaussian forecast uncertainties. If nonlinear observation operators are applied on the expanded ensemble, this contradiction suggests that the expanded ensemble's observation statistics will be biased relative to the true forecasted observation statistics (personal communication with Craig Bishop and Lili Lei).

The shortcomings of existing ensemble expansion methods motivate the development of a new ensemble expansion method. This study proposes an new ensemble expansion method that explicitly utilizes the users' knowledge of prior marginals: the Probit-space Ensemble Size Expansion for Gaussian Copulas (PESE-GC). PESE-GC constructs virtual members using a generalization of the efficient and scalable Gaussian resampling algorithm of Chan et al. (2020) (henceforth, the CAC2020 algorithm). Unlike existing methods, PESE-GC can efficiently and scalably generate an unlimited number of virtual members

with flow-dependent statistics. Furthermore, the PESE-GC produces virtual members that are consistent with user-specified prior marginals, and can handle multivariate Gaussian distributions and many multivariate non-Gaussian distributions. PESE-GC is applied before running any observation operators for EnsDA, and the analyzed virtual members are discarded before generating the next forecast ensemble (see Fig. 1).

The remainder of this publication is divided into five sections. Section 2 discusses the formulation of PESE-GC, and its

computational complexity and scalability. How PESE-GC can improve EnsDA is discussed and illustrated in Section 3. PESE-GC is then tested with EnsDA using the Lorenz 1996 model in Section 4. Section 5 discusses an important caveat of the PESE-GC method. This publication then ends in Section 6 with a summary and a discussion of avenues for future research.

## 2   Formulation of PESE-GC

This section begins with reviewing the CAC2020 algorithm. Then, the CAC2020 algorithm is generalized to handle arbitrary

piecewise-continuous marginal distributions (i.e., 1D distributions) using probit probability integral (PPI) transforms. Finally, the computational complexity and scalability of PESE-GC is discussed.



# EnsDA workflow with ens. expansion

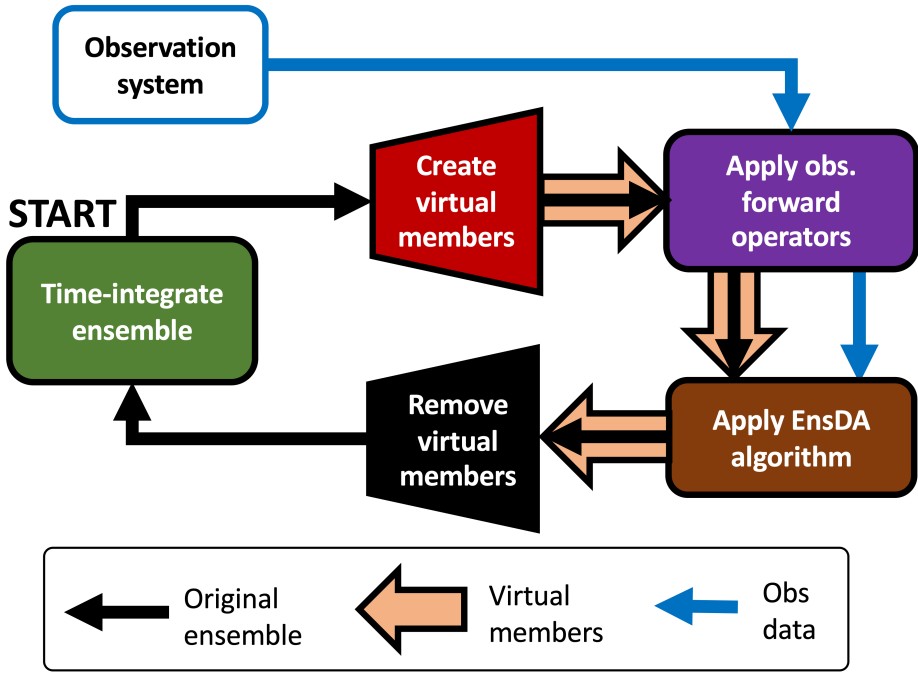

**Figure 1.** An illustration of how PESE-GC can be integrated into a typical ensemble DA (EnsDA) cyclic workflow. This workflow is meant to be read starting from the green box labeled "START". The arrows indicate the movement of various kinds of information (see legend). For example, the fat orange arrows indicate that the virtual members are created by PESE-GC (red polygon), passed to the observation operators (purple rounded box), passed to the EnsDA algorithm (brown rounded box), and then removed before applying the forecast model (black polygon). "Obs" stands for "observation" and "ens." stands for "ensemble".

## 2.1 The CAC2020 algorithm

The CAC2020 algorithm constructs Gaussian-distributed virtual members through linear combinations of the forecast ensemble perturbations. The resulting expanded ensemble has the same mean state and covariance matrix as the forecast ensemble. The CAC2020 algorithm is first formulated by Chan et al. (2020) and a more comprehensive derivation is presented in Chapter 7 of Chan (2022).

To write down the CAC2020 algorithm, a notation similar to Ide et al. (1997) is used. Suppose $\boldsymbol{x}$ is an $N_x$-dimensional column vector representing a forecast model state, $\left\{ \boldsymbol{x}_1^{\boldsymbol{f}}, \boldsymbol{x}_2^{\boldsymbol{f}}, \ldots, \boldsymbol{x}_{\boldsymbol{N_E}}^{\boldsymbol{f}} \right\}$ represents an ensemble of $N_E$ forecast model states,

$$\boldsymbol{x}_{\boldsymbol{n}}^{\boldsymbol{f}\prime} \equiv \boldsymbol{x}_{\boldsymbol{n}}^{\boldsymbol{f}} - \frac{1}{N_E} \sum_{n'=1}^{N_E} \boldsymbol{x}_{\boldsymbol{n'}}^{\boldsymbol{f}}, \ \forall n = 1, 2, \ldots, N_E, \tag{1}$$



$N_V$ virtual members $\left\{x_1^v, x_2^v, \ldots, x_{N_V}^v\right\}$ are constructed, and

$$x_m^{v'} \equiv x_m^v - \frac{1}{N_E} \sum_{m'=1}^{N_V} x_{m'}^v \quad \forall m = 1, 2, \ldots, N_V. \tag{2}$$

Note that the mean of the virtual members is the same as the mean of the forecast members. In other words,

$$\frac{1}{N_V} \sum_{m=1}^{N_V} x_m^v = \frac{1}{N_E} \sum_{n=1}^{N_E} x_n^f. \tag{3}$$

### 2.1.1   Step 1 of the CAC2020 algorithm

The CAC2020 algorithm constructs $N_V$ virtual members from $N_E$ ensemble members using a three-step procedure. First, an $N_E \times N_V$ matrix of linear combination coefficients ($E$) is generated by evaluating

$$E \equiv \gamma \mathbf{1}_{N_E \times N_V} + \mathrm{Chol}\left(C_E\right) \left\{\mathrm{Chol}\left(WW^\top\right)\right\}^{-1} W. \tag{4}$$

Here,

$$\gamma \equiv \frac{1}{N_V} \left( \sqrt{\frac{N_E + N_V - 1}{N_E - 1}} - 1 \right), \tag{5}$$

$\mathbf{1}_{N_E \times N_V}$ is an $N_E \times N_V$ matrix of ones, $\mathrm{Chol}\left(\cdot\right)$ represents Cholesky decomposition, $C_E$ is an $N_E \times N_E$ matrix defined by

$$C_E \equiv \frac{N_V}{N_E - 1} I_{N_E} - \gamma^2 N_V \mathbf{1}_{N_E \times N_E}, \tag{6}$$

$I_{N_E}$ is an $N_E \times N_E$ identity matrix, and $\mathbf{1}_{N_E \times N_E}$ is an $N_E \times N_E$ matrix where every element is one. Finally, $W$ is an $N_E \times N_V$ matrix whose $(i, j)$-th element is defined by

$$W_{i,j} = \omega_{i,j} - \frac{1}{N_V} \sum_{\ell=1}^{N_V} \omega_{i,\ell} \tag{7}$$

where $\omega_{i,j}$ is a random sample drawn from the standard normal distribution.

### 2.1.2   Step 2 of the CAC2020 algorithm

The CAC2020 algorithm's second step is to generate $\left\{x_1^{v'}, x_2^{v'}, \ldots, x_{N_V}^{v'}\right\}$ by evaluating

$$\begin{bmatrix} x_1^{v'} & \cdots & x_{N_V}^{v'} \end{bmatrix} = \begin{bmatrix} x_1^{f'} & \cdots & x_{N_E}^{f'} \end{bmatrix} E. \tag{8}$$

### 2.1.3   Step 3 of the CAC2020 algorithm

In the third and final step, the CAC2020 algorithm generates $\left\{x_1^v, x_2^v, \ldots, x_{N_V}^v\right\}$ by evaluating

$$x_m^v \equiv x_m^{v'} + \frac{1}{N_E} \sum_{n'=1}^{N_E} x_{n'}^f \quad \forall m = 1, 2, \ldots, N_V. \tag{9}$$





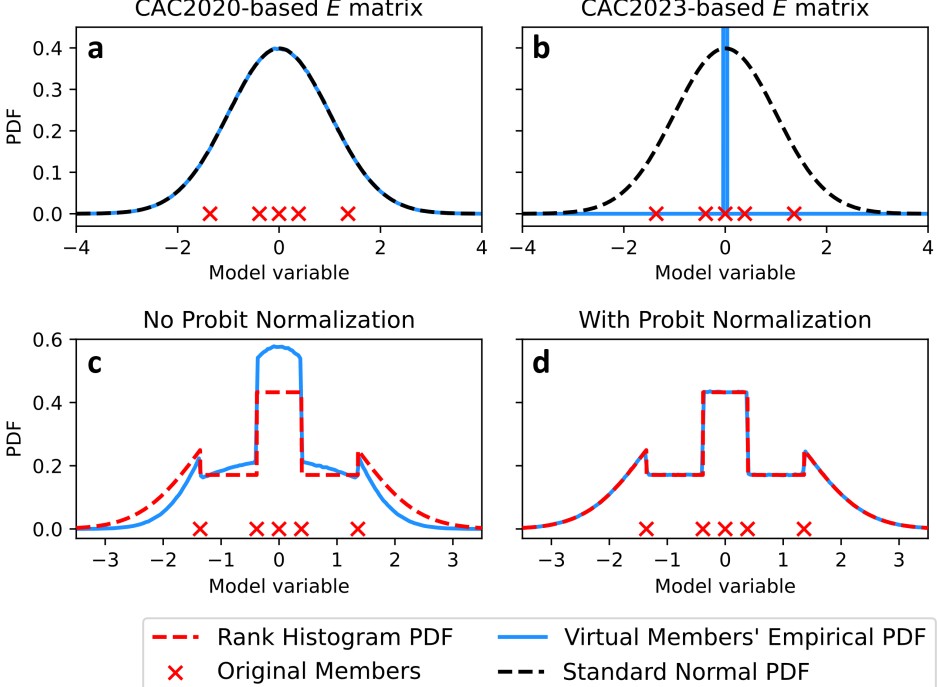

**Figure 2.** Plots demonstrating the impacts of various heuristic choices in the formulation of PESE-GC. Panel a demonstrates using the CAC2020 algorithm to resample 5 forecast members drawn from a standard normal distribution. The resulting virtual members have Gaussian statistics (panel a). In contrast, using the algorithm of Chan et al. (2023) to resample those same 5 members results in virtual members with non-Gaussian statistics (panel b). Panels c and d demonstrate the importance of ensuring that the forecast ensemble in probit-space has unit variance. If the probit-space forecast ensemble's variance is not unity, the virtual members generated by PESE-GC will deviate from the fitted marginal PDFs (rank histogram in the case of panels c and d).

### 2.1.4   On $W$ used in the CAC2020 algorithm's step 1

Note that this study's (and the CAC2020's) $W$ differs from that of Chan et al. (2023) (henceforth, "CAC2023"). This is because the $W$ of Chan et al. (2023) [defined in their Eq. (6)] generates virtual members with undesirable non-Gaussian properties. To illustrate, suppose 5 forecast members are drawn from a standard normal distribution and $10^7$ virtual members are generated using the CAC2020 algorithm with the $W$ of Chan et al. (2023). Though the expanded ensemble's mean and variance are correct (zero and unity respectively), the virtual members' histogram-estimated PDF (blue curve in Fig. 2b) is incorrect (not standard normal). In contrast, using the $W$ defined in Eq. (7) results in virtual members that follow the standard normal PDF (Fig. 2a).





### 2.1.5 The CAC2020 algorithm is efficient and scalable

The CAC2020 algorithm is efficient and scales well with parallel computing. To understand why, note that steps 2 and 3 of the CAC2020 algorithm (Sections 2.1.2 and 2.1.3) are embarrassingly parallel and has a computational complexity that scales linearly with $N_x$. If $\boldsymbol{E}$ is generated offline (i.e., not part of the EnsDA procedure) and then read into the EnsDA program, the CAC2020 algorithm is reduced to only evaluating steps 2 and 3.

### 2.2 The PESE-GC procedure

The CAC2020 algorithm is limited to generating Gaussian-distributed virtual members. PESE-GC overcomes this limitation by combining probit probability integral (PPI) transforms and their inverses with the CAC2020 algorithm. A PPI transform transforms any univariate distribution with a continuous CDF into a standard normal distribution, and the inverse PPI transform reverses the process. The quantity resulting from applying the PPI transform on a random variable is called "probit" and the coordinate space occupied by probits is called "probit-space". Such transforms are often used in Gaussian anamorphosis (Amezcua and Van Leeuwen, 2014; Grooms, 2022).

To define the PPI transform, suppose $F_i(x_i)$ represents the CDF of the $i$-th model variable $x_i$ ($i = 1, 2, \ldots, N_x$), $\Phi^{-1}(q)$ represents the inverse CDF of the standard normal ($q$ represents any quantile), and $\phi_i$ represents the $i$-th model probit. Note that $\Phi^{-1}(q)$ is sometimes called "the probit function" or the "quantile function of the standard normal". The conversion from $x_i$ to $\phi_i$ (i.e., the PPI transform) is

$$\phi_i \equiv \Phi^{-1}\left(F_i(x_i)\right). \tag{10}$$

The inverse PPI transform (i.e., converting $\phi_i$ to $x_i$) is

$$x_i \equiv F_i^{-1}\left(\Phi(\phi_i)\right). \tag{11}$$

The PPI transform generalizes the CAC2020 algorithm to handle non-Gaussian forecast ensembles. The resulting PESE-GC procedure has four stages and is illustrated in Fig. 3. These four stages are:

1. For each model state variable, fit a user-specified univariate distribution to that model variable in the forecast ensemble (i.e., marginal distribution fitting).

2. For each model state variable, apply the PPI transform [Eq. (10)] using that variable's fitted distribution to convert the forecast ensemble of that model variable into forecast probits.

3. For each model state variable, adjust the mean and variance of that variable's forecast ensemble probits to zero and unity, respectively (explained in section 2.3), and then apply the CAC2020 algorithm on that variable's forecast probits to generate virtual probits.

4. For each model state variable, apply the inverse PPI transform [Eq. (11)] with stage 1's fitted distribution on that variable's virtual probits to generate that variable's virtual ensemble.




**Figure 3.** Illustrations of PESE-GC's 4-stage algorithm. Panels a, b, c, and d respectively show the aftermath of stages 1, 2, 3, and 4. The details of these stages are described in Section 2.2.

125      PESE-GC's four-stage procedure is attractive for geophysical EnsDA for several reasons. Aside from the fact that it can generate non-Gaussian virtual members, PESE-GC can be implemented in an embarrassingly parallel fashion (every loop



over the model state variables is embarrassingly parallel). Furthermore, PESE-GC is likely affordable for geophysical EnsDA because the CAC2020 algorithm (stage 3) is efficient (see section 2.1).

Note that the quality of the virtual members depends on the distributions the user selects in step 1 of PESE-GC. This will be discussed later in Section 3.

## 2.3 On the mean and variance adjustments in PESE-GC's step 3

PESE-GC requires forecast ensemble probits with zero mean and unity variance. Otherwise, the resulting virtual members will disobey the marginal distributions fitted in PESE-GC's step 1. To illustrate, suppose PESE-GC is applied on 5 univariate forecast ensemble members (red crosses in Fig. 2c) and a Gaussian-tailed rank histogram distribution (Anderson, 2010) is fitted to those 5 members in PESE-GC's step 1. Applying the PPI transform (PESE-GC's step 2) results in forecast probits with mean zero and a variance of approximately 0.561. $10^7$ virtual probits are then generated from these forecast probits using the CAC2020 algorithm, and the inverse PPI transform is applied to generate the virtual members (PESE-GC's step 4). The histogram-estimated PDF of the virtual members (blue curve in Fig. 2c) disagrees with the fitted (i.e., desired) Gaussian-tailed rank histogram PDF.

This problematic disagreement is resolved by adjusting the forecast probits' mean and variance to zero and unity (respectively) before generating the virtual members. Suppose the probit of the $n$-th forecast member for model variable $i$ is $\phi_{i,n}$ and the pre-adjusted prior ensemble probit's sample mean and sample variance are $\mu$ and $\sigma^2$ respectively. The adjustment process is simply

$$\phi_{i,n} \leftarrow \frac{\phi_{i,n} - \mu}{\sigma} \quad \forall n = 1, 2, \ldots, N_E. \tag{12}$$

The impact of this adjustment is illustrated in Fig. 2: the virtual members' histogram-estimated PDF (blue curve in Fig. 2d) now matches the Gaussian-tailed rank histogram PDF in PESE-GC's step 1.

## 3 Conceptual exploration of PESE-GC's EnsDA impacts

To conceptually understand PESE-GC's impact on EnsDA, consider EnsDA as a two-step Bayesian inference procedure (Anderson, 2003; Grooms, 2022). Suppose an ensemble of forecast observations is obtained by applying observation operators on the forecast ensemble. In the first step, Bayes' rule is used to combine an ensemble of forecast observations with an observation likelihood function. This produces an ensemble of observation increments. Note that an approximate representation of the observation likelihood function is used in some EnsDA methods. For example, the rank histogram filter of Anderson (2010) utilizes a piecewise approximation of the actual observation likelihood function. Another example is the perturbed observation ensemble Kalman Filter (EnKF; Burgers et al. (1998)) – the observation likelihood is represented using random draws from a Gaussian distribution.

In the second step, the ensemble of observation increments is converted into model states increments using the statistical relationships linking forecast model variables and forecast observations. This conversion can be done via regression (Ander-





son, 2003; Anderson et al., 2009), marginal adjustment (Anderson, 2020), or resampling (Doucet, 1998; van Leeuwen, 2009; Poterjoy and Anderson, 2016).

160 The two-step Bayesian inference framework reveals three components that influence the performance of EnsDA:

– the accuracy of the approximate representation of the observation likelihood function (component A), if such approximation is used in the first step,

– the accuracy of the ensemble-sampled relationships linking model variables and observed quantities (component B) that is used in the second step, and,

165 – the accuracy of the forecast observations' sample statistics (component C) used in the first step.

PESE-GC's impact on EnsDA can thus be explored using these three components.

PESE-GC improves all three components if the user has some knowledge of the forecast marginals. This will be illustrated with a bivariate example. Suppose a scalar forecast model variable $x$ has a skewed-normal forecast distribution, 5 samples are drawn from this distribution, a signed square-root function $h(x)$

170 $$h(x) \equiv \text{sign}(x)\sqrt{|x|} \tag{13}$$

will be used as the observation operator, and let $y$ denote observation values. 10,000 virtual members will be generated by PESE-GC in this bivariate example. Note that the true forecast distribution of $x$ is the previously mentioned skewed-normal distribution, and the true forecast distribution of $h(x)$ is estimated by applying $h(x)$ on 1,000,000 samples of $x$ drawn from the true forecast distribution of $x$.

## 3.1 Impact mechanism 1: Better representation of observation likelihood function

PESE-GC improves approximate representations of the observation likelihood function (component A), if such representations are used. For the perturbed observation Ensemble Kalman Filter (stochastic EnKF; Burgers et al. (1998)), PESE-GC increases the number of draws from the observation likelihood function, thus mitigating sampling errors in representing the observation likelihood function Whitaker and Hamill (2002). PESE-GC also improves the piecewise approximation to the observation

180 likelihood function that is used by the rank histogram filter (Anderson, 2010) (illustrated in Fig. 4b). For particle filters (Gordon et al., 1993; van Leeuwen et al., 2019), PESE-GC increases the sampling of the observation likelihood function, potentially delaying the collapse of the particle filter in very large scale systems (Bickel et al., 2008; Bengtsson et al., 2008; Snyder et al., 2008).

## 3.2 Impact mechanism 2: Improved sampling of nonlinear observation operators

185 If the user specifies appropriate marginals in PESE-GC's stage 1, PESE-GC improves component B for nonlinear observation operators. To demonstrate, suppose the user knows that the prior marginal distribution of $x$ is close to Gaussian. Thus, a Gaussian distribution is fitted to the forecast $x$ values in PESE-GC's stage 1 (Section 2.2; Fig. 5a). Applying PESE-GC generates





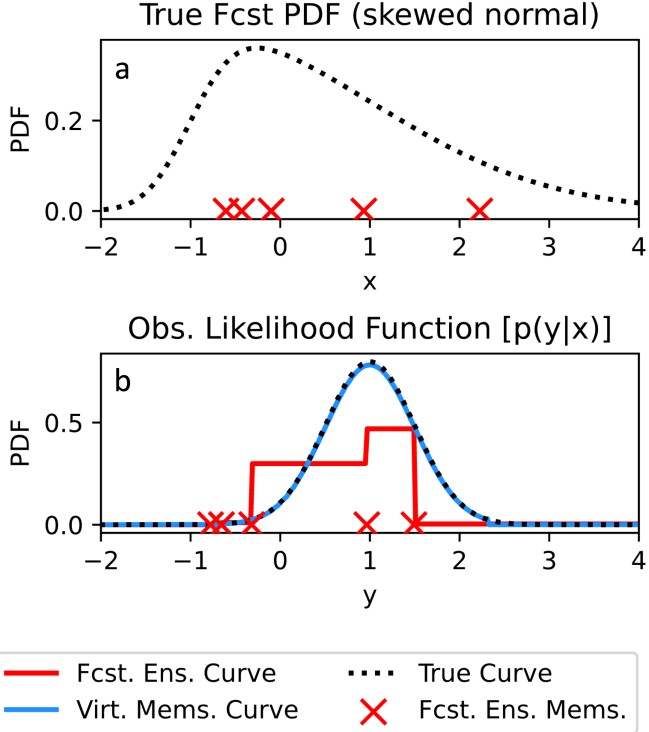

**Figure 4.** Plots of the true forecast PDF (a) and observation likelihood functions (b). The 5 forecast ensemble $x$ values are indicated with red crosses in (a), and the 5 forecast ensemble $y$ values are indicated with red crosses in (b). The red curve in (b) indicates the piecewise-approximated observation likelihood function used by a rank histogram filter that only uses the 5 forecast members. With PESE-GC, the rank histogram filter's piecewise-approximated observation likelihood function (blue curve) is improved.

10,000 virtual members. With more ensemble members (specifically, more unique $x$ values), the nonlinear $h(x)$ is evaluated more often (i.e., better sampled). This increased evaluation improves component B (Fig. 5b), which can ultimately improve EnsDA.

### 3.3 Impact mechanism 3: Improved observation space ensemble statistics

Using appropriate marginals with PESE-GC also improves ensemble statistics in observation space (i.e., component C). This improvement results from both 1) increased sampling of the observation operator, and 2) improved model-space ensemble statistics. To illustrate, suppose the previous paragraph's example is reused. Fig. 5c indicates that the expanded ensemble's $y$ CDF is better than that of the forecast ensemble's. As such, PESE-GC can improve EnsDA through improving component C.



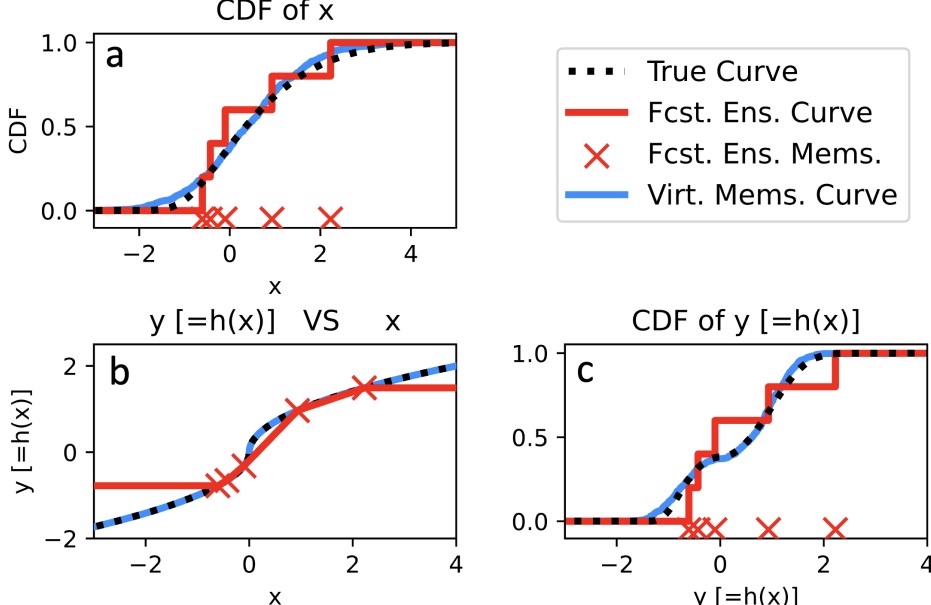

**Figure 5.** Bivariate example demonstrating the impacts of drawing virtual members from an informative fitted marginal (normal distribution). Panel a shows the empirical CDFs of $x$ from the initial members and virtual members. The estimated relationships between $x$ and $y$ (obtained by passing the initial members and virtual members through $h(x)$) are displayed in panel b. Finally, panel c shows the empirical CDFs of the initial members and virtual members for variable $y$. The true CDFs and $x$-$y$ relationships are plotted in panels b and c with dashed black lines.

### 3.4 Caveat: PESE-GC with misinformed marginal distributions may degrade EnsDA's performance

Note that if misinformed marginal distributions are used, PESE-GC may degrade component C, thus degrading the performance of EnsDA. To illustrate, suppose the user fits a shifted gamma distribution (Cheng and Amin, 1983) to the 5 forecast $x$ values. This distribution has three parameters: shape, scale and location. Since only 5 forecast values are used to fit three parameters, the fitted parameters' sampling errors are severe. Applying PESE-GC with this badly estimated shifted gamma distribution results in virtual $x$ statistics that are worse than those of 5 forecast $x$ values (Fig. 6a). Even though component B is slightly improved by PESE-GC (Fig. 6b), the virtual members' $y$ CDF is worse than the forecast members' $y$ CDF (Fig. 6c). This worsening can degrade the performance of EnsDA. As such, the selection of marginals to use with PESE-GC must be done with care.

In the absence of knowledge about the model variables' prior marginal distribution, users can use non-parametric marginal distributions with PESE-GC. Such distributions include the Gaussian-tailed rank histogram distribution (Anderson, 2010) and kernel distibutions (Anderson and Anderson, 1999). While this choice may not improve the model-space ensemble statistics, PESE-GC can still improve components A and B. As such, using PESE-GC with non-parameteric marginal distributions can improve the performance of EnsDA.





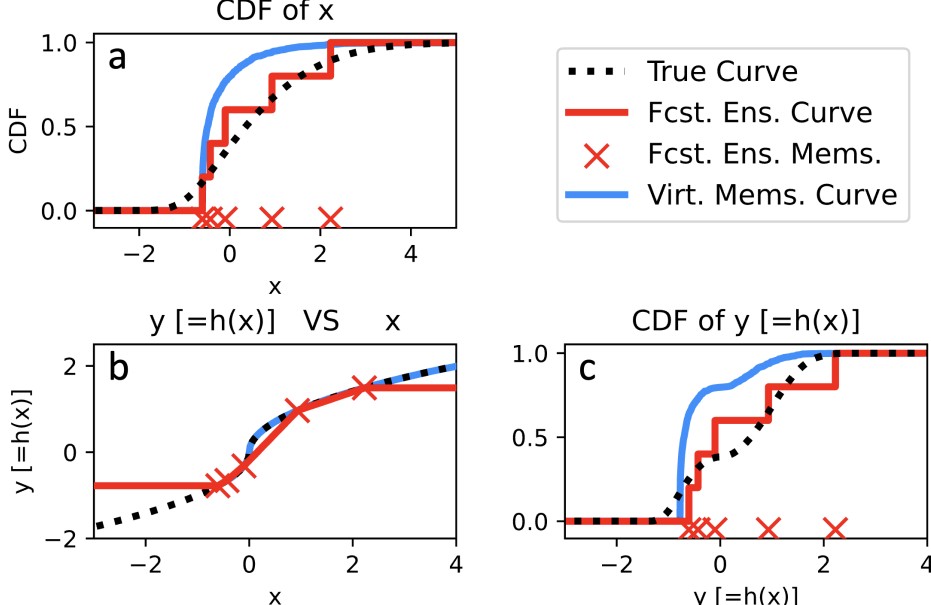

**Figure 6.** Bivariate example demonstrating the impacts of drawing virtual members from a misinformed fitted marginal (gamma distribution). The panels here are similar to Fig. 5.

# 4 Tests with Lorenz 1996 model

## 4.1 Setup of experiments

In the previous section, the mechanisms for PESE-GC to improve the components of EnsDA are examined. In this section, the impacts of PESE-GC on the performance of EnsDA are explored using perfect model Observing System Simulation Experiments (OSSEs) with the Lorenz 1996 model (L96 model; Lorenz (2006)). This exploration is done with the Data Assimilation Research Testbed (DART; `https://github.com/NCAR/DART`; Anderson et al. (2009)). PESE-GC has been implemented into DART.

This study uses DART's implementation of the L96 model with 40 variables (i.e., 40 grid points in a ring), a forcing parameter value of 8 (i.e., $F = 8$), and a time-step of 0.05 L96 time units. This time step is usually taken to be equal to 1 hour. Forward time integration of the model is done via the Runge-Kutta fourth-order integration scheme. Every OSSE experiment runs for 5,500 cycles. Initial nature run states and the initial ensemble members are drawn from the L96's climatology.

In all experiments, there are 40 observations. Their observation locations are fixed throughout this study. Supposing that the model grid points have locations $0.025m$, $m = 1, 2, \ldots, 40$, each site location is a random draw from a uniform distribution between 0 and 1.

PESE-GC's impacts are examined using EnsDA experiments that are conducted with four $N_E$s, five cycling intervals, three post-PESE-GC ensemble sizes, three observation types, four EnsDA algorithms, and with and without PESE-GC, a total of



$4 \times 5 \times 3 \times 3 \times 4 \times 2 = 1440$ configurations. The four $N_E$s are 10, 20, 40, and 80, and the five cycling intervals are 1 hour, 2 hours, 3 hours, 6 hours and 12 hours.

The PESE-GC-expanded ensemble sizes are specified in terms of factors: 5, 10 and 20 times the forecast ensemble size. For example, if $N_E = 10$, a PESE-GC factor of 10 means that the expanded ensemble has 100 members (i.e., $N_V = 90$ virtual members are created).

Supposing the $k$-th observation site has location $\ell_k$, the observation operators for the three observation types are:

$$h_{IDEN}\left(\boldsymbol{x};\ell_k\right) \equiv x_{\ell_k}, \tag{14}$$

$$h_{SQRT}\left(\boldsymbol{x};\ell_k\right) \equiv sgn\left(x_{\ell_k}\right)\sqrt{|x_{\ell_k}|}, \tag{15}$$

$$h_{SQUARE}\left(\boldsymbol{x};\ell_k\right) \equiv sgn\left(x_{\ell_k}\right)\left(x_{\ell_k}\right)^2 \tag{16}$$

where $\boldsymbol{x}$ is the L96 model's 40-element state vector, $x_{\ell_k}$ is the model variable interpolated to location $\ell_k$, and $sgn\left(x_{\ell_k}\right)$ returns the sign of $x_{\ell_k}$. Observations created using $h_{IDEN}\left(\boldsymbol{x};\ell_k\right)$, $h_{SQRT}\left(\boldsymbol{x};\ell_k\right)$ and $h_{SQUARE}\left(\boldsymbol{x};\ell_k\right)$ will be respectively referred to as IDEN observations, SQRT observations and SQUARE observations. Every observation is created by applying its corresponding observation operator on a nature run state, and then perturbing the output with a random draw from $\mathcal{N}\left(0,\sigma^2\right)$. For IDEN, SQRT and SQUARE observations, $\sigma^2$ is 1.0, 0.25, and 16, respectively.

The four EnsDA algorithms tested with PESE-GC are:

1. the Ensemble Adjustment Kalman Filter (EAKF; Anderson (2003))

2. the stochastic EnKF with sorted observation increments (EnKF; Burgers et al. (1998))

3. the Rank Histogram Filter with linear regression (RHF; Anderson (2010))

4. the Rank Histogram Filter with probit regression (PR; Anderson (2023))

To be clear, the RHF algorithm first employs the rank histogram filter to generate observation increments and then uses linear regression to convert observation increments into model increments. The PR algorithm is similar to the RHF algorithm, except that probit regression is used to convert observation increments into model increments.

For each EnsDA algorithm, only 1 set of marginals are used with PESE-GC. When PESE-GC is used with the EAKF, EnKF or RHF algorithm, Gaussian marginals are selected for all 40 model variables. PESE-GC with Gaussian marginals is identical to the CAC2020 algorithm. For the PR algorithm, the Gaussian-tailed rank histogram is selected as the marginal for every one of the 40 model variables. Future work can investigate the impacts of using PESE-GC with Gaussian-tailed rank histograms (or kernel density estimates) with the EAKF, EnKF and RHF.

Each of the 1440 configurations is trialed 36 times. These trials are enumerated (Trial 1, Trial 2, and so forth). All experiments with the same trial number and $N_E$ share the same nature run and initial forecast ensemble states. For example, the following two EnsDA experiments have the same initial nature run and initial forecast ensemble: 1) Trial 10 using IDEN observations, 20 forecast ensemble members, 3 hour cycling period, EAKF, and PESE-GC with an expansion factor of 20, and 2) Trial 10 using





SQRT observations, 20 forecast ensemble members, 12 hour cycling period, RHF, and PESE-GC with an expansion factor of 5. Note that experiments with different trial numbers have different nature runs and initial forecast ensemble states.

The Gaspari-Cohn fifth-order polynomial (Gaspari and Cohn, 1999) is used to localize EnsDA increments. For each combination of trial and configuration, 17 localization half-radii are tested: $0.075 \times 1.3^0, 0.075 \times 1.3^1, 0.075 \times 1.3^2, 0.075 \times 1.3^3, \ldots, 0.075 \times 1.3^{14}, 0.075 \times 1.3^{15}$, and infinity. To select the optimal localization half-radius for a given combination of trial and configuration, the root-mean-square error (RMSE) of the forecast ensemble mean is used. The RMSE for a particular cycle is

$$\mathrm{RMSE} \equiv \sqrt{\frac{1}{40} \sum_{i=1}^{40} \left( \overline{x_i^f} - x_i^t \right)^2} \tag{17}$$

where $\overline{x_i^f}$ and $x_i^t$ are, respectively, the forecast ensemble mean state and nature run state at model grid point $i$. The RMSE of the first 500 out of 5,500 EnsDA cycles are discarded. The localization half-radius that results in the smallest cycle-averaged RMSE (i.e., averaged from cycles 501 to 5,500) is selected as the optimal localization half-radius. Note that for the same configuration, the optimal localization half-radius can vary with the trial number.

The inflation scheme used here is identical to the one used by Anderson (2023). The adaptive prior inflation algorithm of

Anderson (2009) is used with an inflation lower bound of 1 (no deflation), an upper bound of 2, a fixed inflation standard deviation of 0.6, and an inflation damping of 0.9. While manually tuning a homogeneous inflation factor or a relaxation-to-posterior-spread (RTPS; **?**) relaxation factor may give smaller RMSEs, an adaptive inflation approach is chosen to reduce the computational cost of this study. This study already runs 881,280 ($1,440 \times 36 \times 17$) combinations of configurations (1,440), trials (36), and localization half-radii (17), and each combination is run for 5,500 cycles.

**4.2   Metric to assess PESE-GC's impact on EnsDA**

The impacts of PESE-GC on EnsDA are assessed using the relative difference between cycle-averaged RMSEs (Eq. (17)) with PESE-GC and cycle-averaged RMSEs without PESE-GC. The cycle averaging is done from cycles 501 to 5,500. To define this RMSE relative difference, suppose an arbitrary configuration of $N_E$, cycling interval, PESE-GC factor, observation type, and EnsDA algorithm is denoted by $\boldsymbol{\xi}$. Let $\overline{\mathrm{RMSE}\,(\boldsymbol{\xi}, r, \mathrm{True})}$ denote the cycle-averaged RMSE of the $r$-th trial of configuration $\boldsymbol{\xi}$

with PESE-GC used. Furthermore, let $\overline{\mathrm{RMSE}\,(\boldsymbol{\xi}, r, \mathrm{False})}$ denote the cycle-averaged RMSE of the $r$-th trial of configuration $\boldsymbol{\xi}$ without using PESE. The relative difference between $\overline{\mathrm{RMSE}\,(\boldsymbol{\xi}, r, \mathrm{True})}$ and $\overline{\mathrm{RMSE}\,(\boldsymbol{\xi}, r, \mathrm{False})}$ is defined as

$$\Delta \mathrm{RMSE}\,(\boldsymbol{\xi}, r) \equiv \frac{\mathrm{RMSE}\,(\boldsymbol{\xi}, r, \mathrm{True}) - \mathrm{RMSE}\,(\boldsymbol{\xi}, r, \mathrm{False})}{\left\langle \overline{\mathrm{RMSE}\,(\boldsymbol{\xi}, \mathrm{False})} \right\rangle}. \tag{18}$$

Here, the denominator is the trial-averaged (indicated by angled brackets) cycle-averaged RMSE of EnsDA run with the same $\boldsymbol{\xi}$ as in the numerator, but with PESE-GC unused. For readability, $(\boldsymbol{\xi}, r)$ is omitted from the rest of this paper. Most importantly,

a negative $\Delta\mathrm{RMSE}$ indicates that PESE-GC improves the performance of EnsDA, and a positive value of $\Delta\mathrm{RMSE}$ indicates that PESE-GC degrades the performance of EnsDA.




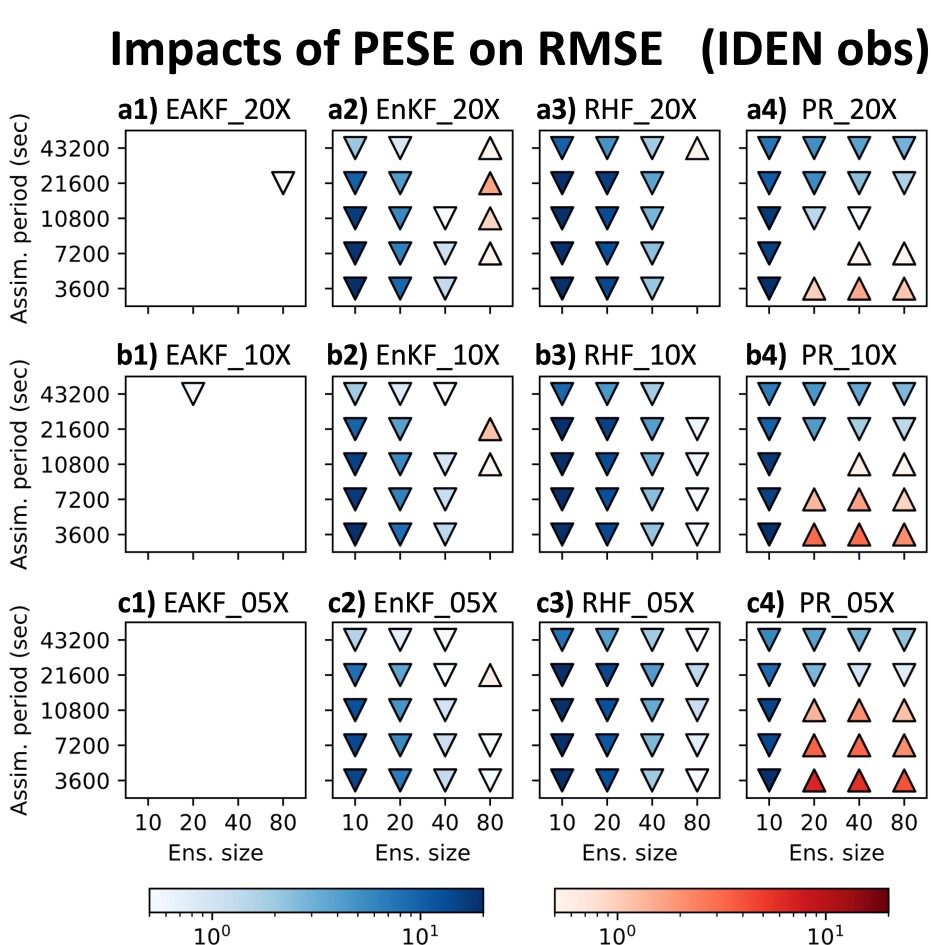

**Figure 7.** Statistically significant $\langle \Delta\text{RMSE} \rangle$ values (two-tailed $p < 0.01$) for pairs of PESE-GC-using and PESE-GC-omitting experiments that assimilate IDEN observations. The PESE-GC-using EnsDA experiments in panels a1, a2, a3 & a4 expanded their forecast ensembles 20-fold, the PESE-GC-using EnsDA experiments in panels b1, b2, b3 & b4 expanded their forecast ensembles 10-fold, and the PESE-GC-using EnsDA experiments in panels c1, c2, c3 & c4 expanded their forecast ensembles 5-fold. Panels a1, b1 and c1 show $\langle \Delta\text{RMSE} \rangle$ for EnsDA experiments using the EAKF, panels a2, b2 and c2 show $\langle \Delta\text{RMSE} \rangle$ for EnsDA experiments using the EnKF, panels a3, b3 and c3 show $\langle \Delta\text{RMSE} \rangle$ for EnsDA experiments using the RHF, and panels a4, b4 and c4 show $\langle \Delta\text{RMSE} \rangle$ for EnsDA experiments using the PR. "Assim. period" is synonymous with "cycling interval", and "ens. size" is synonymous with "$N_E$". The acronyms are defined in Section 4.1. Downward triangles indicate $\langle \Delta\text{RMSE} \rangle < 0$ and upward triangles indicate $\langle \Delta\text{RMSE} \rangle > 0$.

Only statistically significant trial-averaged $\Delta\text{RMSE}$ (henceforth, $\langle \Delta\text{RMSE} \rangle$) will be discussed in this paper. A $\langle \Delta\text{RMSE} \rangle$ is considered statistically significant if its two-tailed z-test p value is smaller than 1%. These statistically significant $\langle \Delta\text{RMSE} \rangle$ values are plotted in Figs. 7, 8, and 9.





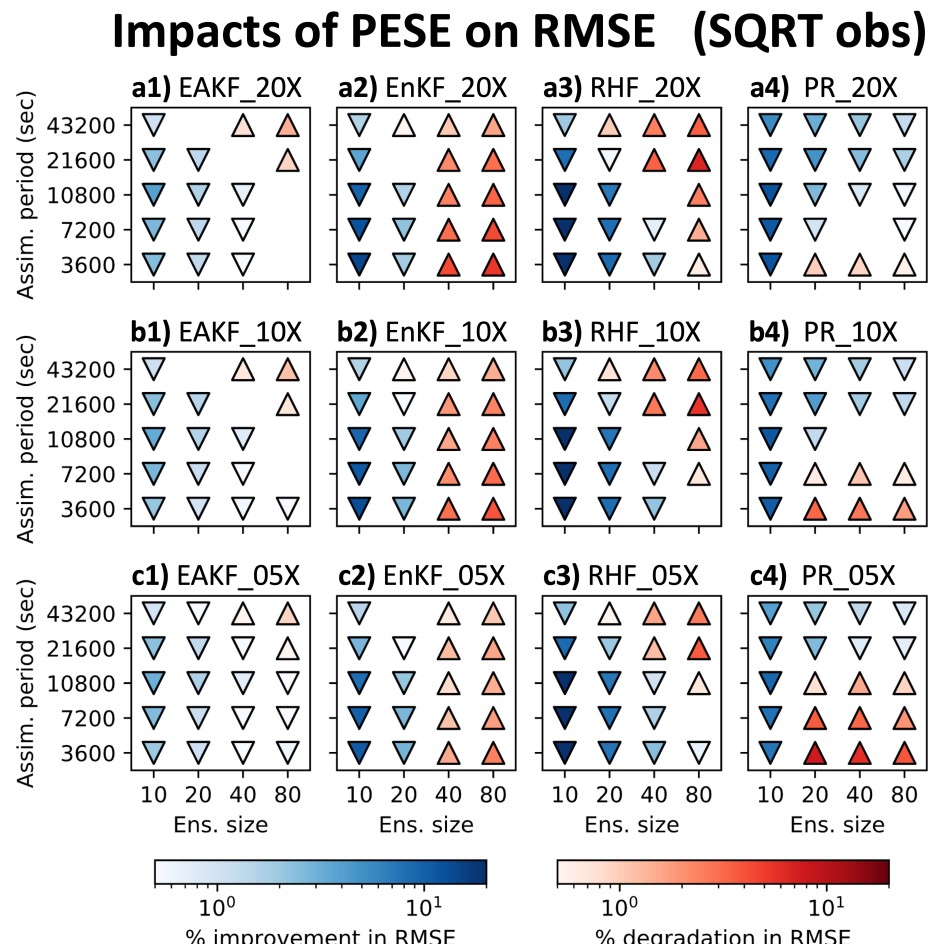

**Figure 8.** Similar to Fig. 7, except that SQRT observations are assimilated.

### 4.3 PESE-GC has negligible impact on EAKF with IDEN observations

Before proceeding, note that PESE-GC with Gaussian marginals only negligibly changes the performance of the EAKF with
IDEN observations (Fig. 7a1, b1 & c1). This negligibility is because none of the three mechanisms listed in Section 3 are
active. Since the EAKF does not represent the observation likelihood function via random draws or a finite number of eval-
uations, impact mechanism 1 of Section 3 does not apply. Furthermore, because PESE-GC is used with Gaussian marginals,
the expanded ensemble has the same mean and covariance as the forecast ensemble. For deterministic and linear observation
operators (e.g., IDEN), this sameness prevents PESE-GC from altering the linearly regressed relationship between forecast
observation values and forecast model values (i.e., impact mechanism 2 of Section 3 does not apply). This also means the
mean and covariance of the forecast observations are unchanged by PESE-GC. As such, any changes introduced by PESE-GC





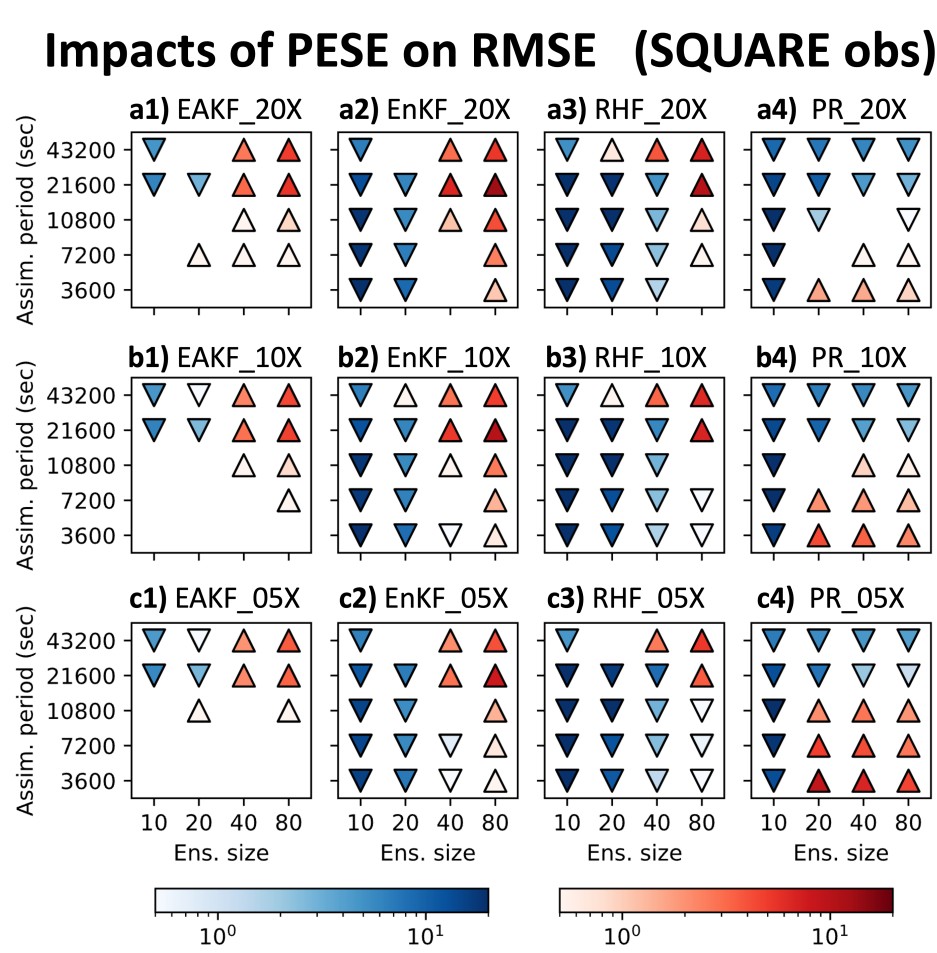

**Figure 9.** Similar to Fig. 7, except that SQUARE observations are assimilated.

in the EAKF experiments with IDEN observation are entirely due to floating point precision errors. The impact of PESE-GC

on the EAKF experiments with IDEN observations will be omitted from subsequent discussion.

### 4.4 Impacts of using 20-fold PESE-GC on EnsDA

This study first examines the $\langle\Delta\text{RMSE}\rangle$ for a PESE-GC factor of 20 (panels a1, a2, a3 and a4 in Figs. 7, 8 and 9). The focus is
on identifying common patterns in $\langle\Delta\text{RMSE}\rangle$ and explaining them through the lens of the three impact mechanisms laid out in
Section 3. The variation of $\langle\Delta\text{RMSE}\rangle$ with different PESE-GC factors (remaining panels in Figs. 7, 8, and 9) will be discussed

in Section 4.5.

The first common $\langle\Delta\text{RMSE}\rangle$ pattern in the 20-fold PESE-GC situations is that PESE-GC generally improves EnsDA per-
formance (i.e., $\langle\Delta\text{RMSE}\rangle < 0$) when $N_E$ is 10 or 20 (panels a1, a2, a3 and a4 in Figs. 7, 8 and 9). This is because either one,





two or all three impact mechanisms described in Section 3 are acting to improve RMSEs. First, for the PR, RHF and EnKF experiments, the performance improvements are partly (if not entirely) due to impact mechanism 1 (Section 3.1). For the PR

experiments with IDEN observations, impact mechanism 1 is likely the sole reason for the improved performance. Second, the PESE-GC-induced RMSE reductions in all 10/20 forecast member experiments with either SQRT or SQUARE observations are partly due to improved sampling of the observation operator (i.e., impact mechanism 2 described in Section 3.2).

The RMSE reductions seen in the 10/20-member EAKF, EnKF and RHF experiments are also partly due to impact mechanism 3 (Section 3.3). This is because the L96 model's forecast statistics tend to be close to Gaussian (e.g., Chan et al. (2020)).

Applying PESE-GC with Gaussian marginals thus improves the model variables' prior ensemble statistics, therefore activating impact mechanism 3.

The second common pattern in the 20-fold PESE-GC experiments is that with increasing $N_E$, PESE-GC's RMSE impacts can go from improving ($\langle \Delta \mathrm{RMSE} \rangle < 0$) to degrading ($\langle \Delta \mathrm{RMSE} \rangle > 0$). This pattern is likely related to impact mechanisms 1 and 3 of Section 3. First, for the EnKF, RHF and PR, the representation of the observation likelihood function improves with

increasing $N_E$. This improvement implies there is less room for PESE-GC to improve that representation. As such, impact mechanism 1 weakens with increasing $N_E$, thus reducing PESE-GC-induced EnsDA performance gains for the EnKF, RHF and PR EnsDA algorithms.

For the EAKF, EnKF and RHF experiments, impact mechanism 3 also contributes to the worsening of PESE-GC's RMSE impacts with increasing $N_E$. In the act of choosing Gaussian marginal distributions for PESE-GC, the user implicitly assumes

that the true forecast PDF is Gaussian. With increasing $N_E$, imperfections in this Gaussian assumption become increasingly evident. The impacts of PESE-GC on the observation space prior statistics can thus go from improving to degrading with increasing $N_E$. This change likely contributes to the worsening of PESE-GC's RMSE impacts with increasing $N_E$.

A third common pattern is that with longer cycles, the PESE-GC's RMSE impacts on the EAKF, EnKF and RHF degrades (i.e., $\langle \Delta \mathrm{RMSE} \rangle$ goes from either negative to zero, zero to positive, or negative to positive). This pattern is likely due to

increasing non-Gaussianity in the forecast ensemble's statistics with longer cycles. The choice to use Gaussian marginals in these experiments' PESE-GC thus becomes increasingly inappropriate, meaning that impact mechanism 3 increasingly degrades the performance of EnsDA.

The fourth common pattern is that in the PR experiments, for $N_E \geq 20$, the 20-fold PESE-GC's impact improves with longer cycling interval (Figs. 7a4, 8a4 and 9a4). A plausible explanation relates to PR's usage of a piecewise approximation to the

observation likelihood function (henceforth, the "piecewise approximation"). This approximation is more accurate when more ensemble members sample the regions of the observation likelihood function where that function varies strongly. However, with increasing forecast ensemble variance, those regions tend to be less sampled by forecast ensemble members, thus degrading the piecewise approximation. Since longer cycling intervals increase forecast ensemble variance, longer cycling intervals thus increase the room for PESE-GC to improve the piecewise approximation. Future work can test this explanation.

Note that the chain of events discussed in the previous paragraph likely occurs for the RHF experiments as well. Since the RHF experiments do not exhibit the fourth common pattern, it is likely that the inappropriateness of the Gaussian marginals used with PESE-GC overwhelms improvements introduced by refining the piecewise approximation.





### 4.5 Variations in PESE-GC's impacts with different amounts of ensemble expansion

This study now examines common patterns in how PESE-GC's impacts vary with PESE-GC expansion factors. The first
common pattern is that PESE-GC's impacts on the PR experiments tends to weaken with smaller PESE-GC factors (panels a4,
b4 and c4 in Figs. 7, 8 and 9). This pattern is likely caused by increasing sampling errors in the virtual members' statistics with
fewer virtual members (i.e., smaller PESE-GC factors).

The second common pattern is that, for $N_E \geq 40$, smaller PESE-GC factors tend to result in milder RMSE degradations
introduced by PESE-GC in the EAKF, EnKF and RHF experiments. Since these RMSE degradations are likely due to the
misinformed selection of Gaussian marginals with PESE-GC (see section 4.4), reducing the number of virtual members reduces
the amount of misinformation introduced by PESE-GC into the forecast statistics. These less misinformed forecast statistics
explain the second common pattern.

An interesting third common pattern is also visible in the EAKF, EnKF and RHF experiments: there are instances where
reducing PESE-GC factors 1) turns insignificant RMSE impacts into RMSE improvements (e.g., the lower left corner of Figs.
8a1 and c1), or 2) turns RMSE degradations into RMSE improvements (e.g., upper right corner of Figs. 7a3 and c3). To
explain this third pattern, notice that these instances are associated with short cycling intervals (1-2 hours) and shorter cycling
intervals are associated with increasingly Gaussian forecast PDFs. Based on these associations, a plausible explanation is that
the true forecast statistics only mildly deviate from Gaussian, but the forecast ensemble statistics are often far from Gaussian.
Introducing some Gaussian virtual members thus improves the ensemble statistics. However, if too many Gaussian virtual
members are introduced, the expanded ensemble statistics become too close to Gaussian. This "Goldilocks" explanation can
be tested in future work.

Most importantly, even with a mere 5-fold PESE-GC, PESE-GC improves the performance of EnsDA in three types of
situations. First, all EnsDA experiments involving small forecast ensemble sizes (10 members) are improved by PESE-GC.
Second, situations where using Gaussian marginals with PESE-GC improves ensemble statistics are also improved by PESE-
GC. This second type of situation occurs for the EAKF, EnKF and RHF experiments that either have 1) 20-40 ensemble
members and/or 2) cycling intervals that are 6 hours or less. Third, PESE-GC improves the PR experiments for cycling intervals
that are 6 hours or 12 hours. This improvement is plausibly because with longer cycling intervals, PESE-GC better improves
the piecewise observation likelihood approximation used by the PR EnsDA algorithm (explained in Section 4.4). These PESE-
GC-introduced improvements are particularly encouraging because a geophysical EnsDA system is more likely able to afford
using 5-fold PESE-GC over 20-fold PESE-GC.

## 5 Discussion

The results presented in the previous section are encouraging. Furthermore, thanks to the utilization of PPI transforms and
their inverses, PESE-GC can generate virtual members in non-Gaussian forecast situations. However, a caveat about PESE-GC
needs discussion: PESE-GC assumes that the forecast distribution is a multivariate Gaussian distribution in probit space (i.e.,
Gaussian copula). This can be seen from the use of a multivariate Gaussian resampling algorithm (the CAC2020 algorithm) to



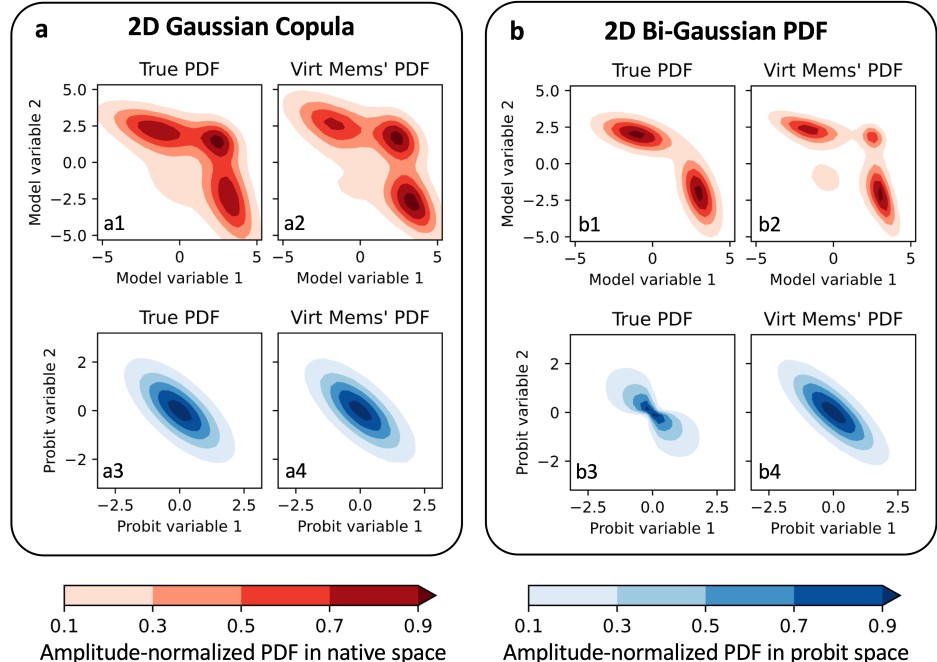

**Figure 10.** Two bivariate demonstrations of PESE-GC. In each demonstration, 100 initial members are drawn from a true bivariate PDF (a1, b1), Bi-Gaussian PDFs are fitted to each variable, and PESE-GC creates 1,000,000 virtual members. Panels a1 and b1 show the PDFs that the initial members are drawn from (i.e., the true bivariate PDFs), panels a2 and b2 show bivariate empirical PDFs estimated by the virtual members, panels a3 and b3 show the true bivariate PDF in probit space, and panels a4 and b4 show the virtual members' bivariate empirical PDFs in probit space. The two true bivariate are: a trimodal PDF with an underlying Gaussian copula (a1) and a bi-Gaussian PDF (panel d1). Note that the bi-Gaussian PDF's copula is not a Gaussian copula (b3). The two true PDFs are described in Section 5.

construct virtual probits. If that assumption is violated (henceforth, "Gaussian copula assumption"), the virtual members will possess statistical artifacts.

Fig. 10a illustrates PESE-GC's ability to generate non-Gaussian virtual members for a situation where the Gaussian copula assumption holds. The true forecast multivariate PDF (Fig. 10a1) is created by applying two inverse PPI transforms on a bivariate Gaussian PDF. The two-dimension mean vector $\boldsymbol{\mu}$ and the $2 \times 2$ covariance matrix $\boldsymbol{\Sigma}$ of the bivariate Gaussian PDF are

$$\boldsymbol{\mu} \equiv \begin{bmatrix} 0 & 0 \end{bmatrix}^{\top}, \, \boldsymbol{\Sigma} \equiv \begin{bmatrix} 1 & 0.7 \\ 0.7 & 1 \end{bmatrix}. \tag{19}$$

The first inverse PPI transform is applied on the first variable ($x_1$) and the PDF it uses ($p(x_1)$) is

$$p(x_1) \equiv \frac{1}{2}G(x_1; -1, 2) + \frac{1}{2}G(x_1; 3, 1) \tag{20}$$





where $G(x_1; -1, 2)$ represents the Gaussian PDF with scalar mean $-1$ and standard deviation 2, and likewise for $G(x_1; 3, 1)$. The second inverse PPI transform is applied on the second variable $(x_2)$ and the PDF it uses $(p(x_2))$ is

$$p(x_2) \equiv \frac{1}{2}G(x_2; 2, 1) + \frac{1}{2}G(x_2; -2, 2) \tag{21}$$

where $G(x_2; 2, 1)$ and $G(x_2; -2, 2)$ are defined similar to $G(x_1; -1, 2)$.

Since the forecast PDF in Fig. 10a1 has, by construction, a Gaussian copula (Fig. 10a3), PESE-GC can produce virtual members that approximately follow the forecast PDF. To demonstrate, 100 forecast members were drawn from the forecast PDF, and 1,000,000 virtual members were constructed using PESE-GC. The two marginal PDFs that are used in steps 1, 2 and 4 of PESE-GC are univariate Bi-Gaussian PDFs (fitted via maximum likelihood estimation in step 1). The histogram-estimated PDFs of the virtual members (Fig. 10a2) and virtual probits (Fig. 10a4) are similar to the true forecast PDF (Fig. 10a1) and the true forecast's probit-space PDF (Fig. 10a3).

An example where the Gaussian copula assumption is violated is shown in Fig. 10b. Here, the forecast PDF (Fig. 10b1) is the following bivariate bi-Gaussian PDF

$$p(\boldsymbol{x}) = \frac{1}{2}G(\boldsymbol{x}; \boldsymbol{\mu_1}, \boldsymbol{\Sigma_1}) + \frac{1}{2}G(\boldsymbol{x}; \boldsymbol{\mu_2}, \boldsymbol{\Sigma_2}) \tag{22}$$

where

$$\boldsymbol{x} \equiv \begin{bmatrix} x_1 \\ x_2 \end{bmatrix}, \ \boldsymbol{\mu_1} \equiv \begin{bmatrix} -1 \\ 2 \end{bmatrix}, \ \boldsymbol{\mu_2} \equiv \begin{bmatrix} 3 \\ -2 \end{bmatrix}, \tag{23}$$

$$\boldsymbol{\Sigma_1} \equiv \begin{bmatrix} 2 & -0.5 \\ -0.5 & 0.5 \end{bmatrix}, \ \boldsymbol{\Sigma_2} \equiv \begin{bmatrix} 0.5 & -0.5 \\ -0.5 & 2 \end{bmatrix}. \tag{24}$$

Applying PPI transforms on this bivariate bi-Gaussian forecast PDF reveals that the bi-Gaussian PDF violates the Gaussian copula assumption (Fig. 10b3). When PESE-GC is applied to generate 1,000,00 virtual members from 100 forecast members, the virtual members' probit space bivariate PDF (Fig. 10b4) differs from the forecast PDF in probit space (Fig. 10b3). As such, the virtual members' bivariate PDF (Fig. 10b2) deviates from the true bivariate bi-Gaussian forecast PDF (Fig. 10b1; the virtual members have two small spurious modes).

Note that though the virtual members' PDF deviates from the forecast PDF, a strong similarity exists between the two PDFs. The two dominant modes of the virtual members' PDF are very similar to the bi-Gaussian forecast PDF. More generally, milder violations of the Gaussian copula assumption will likely lead to milder spurious statistical features in the virtual members.

More importantly, PESE-GC's Gaussian copula assumption may not be problematic for geophysical EnsDA. Due to the high dimensionality of geophysical models and small forecast ensemble sizes, it is difficult to identify the family of the multivariate forecast distributions in probit space. In other words, the forecast ensemble's statistics in probit space are likely indistinguishable from a multivariate Gaussian. This indistinguishability permits assuming Gaussian copulas. Future work can investigate this possibility.





## 6   Summary and future work

In this study, an efficient and embarrassingly parallel algorithm to increase ensemble sizes, PESE-GC, is formulated. PESE-GC generalizes the efficient and embarrassingly parallel Gaussian resampling algorithm of Chan et al. (2020) to handle non-Gaussian forecast distributions. This handling of non-Gaussian forecast distributions means PESE-GC is highly flexible. Furthermore, PESE-GC provides an avenue for users to use their knowledge of the forecast statistics to improve EnsDA – users can choose to draw virtual members using marginal distribution families (e.g., Gaussian and gamma distribution families) that they

think are good approximations to the true forecast marginal distributions. If that knowledge is unavailable, users can choose to use non-parameteric marginal distributions (e.g., Gaussian-tailed rank histogram distributions).

Three mechanisms are then identified for PESE-GC to influence the performance of EnsDA. First, for EnsDA methods like the stochastic EnKF and rank histogram filter, PESE-GC improves the representation of the observation likelihood function. Second, by expanding the number of ensemble members, PESE-GC increases the sampling of the observation operator. This

increased sampling improves the forecast observations' PDF. Finally, when users use PESE-GC with informative marginal distribution families, the forecast observations' statistics are improved.

The impacts of PESE-GC on the performance of EnsDA are explored using the L96 model, a variety of observation systems and a variety of EnsDA algorithms. Results indicate that PESE-GC generally improves the performance of EnsDA when 1) the forecast ensemble size is small (10 members), 2) the marginal distribution families used with PESE-GC are informative, and/or

3) PESE-GC improves the representation of the observation likelihood function (the PR experiments in sections 4.4 and 4.5). It is particularly encouraging that many of these improvements are retained even with a modest amount of ensemble expansion (5-fold expansion).

There are two general areas for future work with PESE-GC. The first area is to move PESE-GC towards geophysical models (EnsDA or forecast postprocessing). To do so, PESE-GC needs to be first tested with ensemble members created by geophysical

models (e.g., Weather Research and Forecasting model; Skamarock et al. (2008)). It will be particularly interesting to see if the virtual members have realistic meteorological structures (e.g., convective clouds with supporting circulations). Then, PESE-GC can be tested using geophysical EnsDA and/or postprocessing system. If PESE-GC does improve the performance of geophysical EnsDA/postprocessing, a comparison between PESE-GC and other ensemble expansion methods is warranted.

Another general area for future work is to develop the PESE-GC algorithm further. First, given the importance of localization

in practical EnsDA, future work can and should explore inserting localization into PESE-GC. Second, the validity of PESE-GC's Gaussian copula assumption can be assessed in the context of geophysical modelling and forecasting. If the Gaussian copula assumption is inappropriate, then non-parametric methods to generate virtual probits can be explored. Finally, methods to detect the usage of misinformed parametric marginal distribution families deserve exploration. One possible detection method is to employ hypothesis testing on the marginal distributions. For example, if Gaussian distributions are selected for

PESE-GC, then the Shapiro-Wilk test can be applied on the forecast ensemble to determine if the selection is misinformed (e.g., Kurosawa and Poterjoy (2023)).

The computational cost of running geophysical models will continue increasing in the coming years (higher spatial resolution, shorter time steps, more complex parameterization schemes, etc). Geophysical EnsDA groups will continue to grapple with the challenge of balancing the computational costs of increasing the number of forecast ensemble members and the computational costs of using more realistic geophysical models. If ensemble expansion methods can provide much of the benefits of a larger forecast ensemble size at a fraction of the cost, these methods will enable EnsDA groups to employ more realistic geophysical models.

*Code availability.* The codes used in this study is publicly available in Chan (2023). These codes include 1) Python scripts used to generate the conceptual illustrations, 2) an implementation of PESE-GC into DART, and 3) Python and Bash scripts used to run and evaluate this study's Lorenz 1996 experiments.

*Author contributions.* The author, Man-Yau Chan, created the PESE-GC algorithm, implemented PESE-GC into DART, designed and executed the experiments discussed in this publication, plotted every figure in this publication, and drafted and edited every version of this publication.

*Competing interests.* The author have no competing interests with this study.

*Disclaimer.* Any opinions, findings, and conclusions or recommendations expressed in this publication are those of the authors and do not necessarily reflect the views of the National Science Foundation.

*Acknowledgements.* The author is eternally grateful to Jeffrey L. Anderson and the National Center for Atmospheric Research's Data Assimilation Research Section for useful discussions and guidance. The author thanks Ian Grooms, Mohamad El Gharamti, and Craig Schwartz for helping to improve the explanation of the PESE-GC algorithm. Furthermore, the author thanks the participants of the International Symposium for Data Assimilation 2023 (ISDA), particularly Alberto Ortolani, for discussions that further clarified the author's thinking process and explanations. Finally, the author thanks Yao Zhu and Christopher Hartman for checking the readability of this manuscript.

This study is supported by the Advanced Study Program Postdoctoral Fellowship at the National Center for Atmospheric Research (NCAR) and the Ohio State University. NCAR is sponsored by the National Science Foundation. All computations in this study are done on two NCAR computing clusters: Casper and Cheyenne. These clusters are managed by NCAR's Computational Information Systems Laboratory.



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
