# Peer review of "Improving Ensemble Data Assimilation through Probit-space Ensemble Size Expansion for Gaussian Copulas (PESE-GC)"

_EGUsphere, 2023_

## Referee Comment (RC2)

Title: Improving Ensemble Data Assimilation through Probit-space Ensemble Size Expansion for Gaussian Copulas (PESE-GC)

Authors: Man-Yau Chan

Recommendation: Minor revision

Summary

This manuscript nicely proposed a method to expand the ensemble size in probit space. Unlike commonly used methods for ensemble size expansion, this PSES-GC ensemble size expansion method can generate virtual members from any multivariate forecast distribution with a Gaussian copula, by using the users' knowledge of the marginal distributions of forecast model variables. This PSES-GC method has been tested with tremendous scenarios, using the L96 model. Results show that PSES-GC can improve the performances of ensemble-based data assimilation methods when the actual ensemble size is small, or the marginal distributions improves the forecast model variable statistics, or the rank histogram filter is used with non-parametric priors. Expanding the ensemble size has advantages to ensemble-based DA in many aspects, while this PSES-GC is an interesting one. The manuscript is well written and pleasant to read. I have several specific comments as below.

1. l28, I'd like to bring up two references here. Sun et al. (2022, 2024) use "analog" ensemble for paleoclimate data assimilation. The paleoclimate data assimilation could be a potential application for the ensemble size expansion, since very limited ensemble members are available.

Sun, H., L. Lei, Z. Liu, L. Ning, and Z.-M. Tan, 2024: A hybrid gain analog offline EnKF for paleoclimate data assimilation. *J. Adv. Model. Earth Syst.,* **16**, e2022MS003414, doi: https://doi.org/10.1029/2022MS003414

Sun, H., L. Lei, Z. Liu, L. Ning, and Z.-M. Tan, 2022: An analog offline EnKF for paleoclimate data assimilation. *J. Adv. Model. Earth Syst.,* **14**, e2021MS002674, doi: https://doi.org/10.1029/2021MS002674

2. l33-34, "However, ensemble modulation assumes that forecast uncertainties have Gaussian statistics." This statement is not right clear to me. Could you explain a little bit why the modulation assumes Gaussian statistics? The modulation just applies localization, not really an ensemble size expansion.

3. l89, it would be nice to have Eq. (6) of Chan et al. (2023) here. Then the reader can immediately see the difference between Eq. (6) of Chan et al. (2023) and the W in this manuscript.

4. Section 2.2 and Figure 3, it is not straightforward to see how the cross-variable relationships are handled? Are the covariances of model state variables naturally conserved by performing PPI for each kind of state variable?

Again at l132, does the adjustment conserve the covariance in model space?

5. l160-165, it is not quite clear the difference between the components A and C. Are they both about $p(y|x)$?

6. l239, any explanation for the choice of the error variance of 0.25 and 16 for the SQRT and SQUARE observations, given 1.0 for IDEN?

7. l272, the reference for RTPS is missing.

8. Section 4.4, here mechanisms 1-3 are used. It would be better to consistently use components A, B and C as in previous discussions.

9. l323-327, I don't quite understand here. I thought it should be more Gaussian if in probit space.

10. l340-342, I am not convinced here. The RHF can be seen as a non-Gaussian filter.

11. It is nice to virtually increase the ensemble size. But it would be nicer to discuss the computational cost for the DA process, with increased ensemble sizes.

---

## Editor Comment (EC1)

The two referees have evaluated positively the paper, and recommend acceptance, subject to minor revisions.

Referee 1 (who has let his name known, and is Ian Grooms) has a major comment about Section 3 of the paper, which he finds a 'bit too vague', and for which he would like clearer explanations. He has in addition a number of minor, mostly editorial, comments.

Referee 2 has also a number of minor comments, which have to do with both scientific and editorial aspects.

I as Editor have also a number of comments (I mention that there seems to be a shift of a few units between the line numbers below and the ones mentioned by the referees).
.

1. I wonder in particular what the virtues of the extended PESE-GC ensemble are. I am sure these virtues are described in the previous publications of the author (in particular Chan *et al.*, 2020, and Chan, 2022), but it could be useful to say a little more in the present paper for readers who are new to that method. From what I understand, specific properties of the extended PESE-GC ensemble are that it has the same mean and covariance matrix as the forecast ensemble, that it preserves through PPI the marginal distributions of the forecast model variables, and that it is numerically very economical. Would any other ensemble expansion method that had the same properties (maybe there is not any) be as useful ? In particular does the CAC2020 algorithm (subsection 2.1) introduce additional properties of interest ? Additional explanations on those points, even succinct, could be useful (this aspect may have to do with Referee 1's major comment).

2. My understanding is that the CAC2020 algorithm is implemented between steps 3 and 4 of the PESE-GC procedure (ll. 122-126). Say it there. And, if I am mistaken, additional appropriate information will be useful.

3. Step 3 of PESE-GC and subsection 2.3. Is the need to adjust the mean and variance of each variable to 0 and 1 due only to the finiteness of the forecast ensemble ? If yes, say it explicitly. If no, explain more clearly.

And for a number of editing comments

4. Eq. (2), denominator on the rhs $N_E \rightarrow N_V$

5. Eq. (4). What does Chol(.) exactly mean here ? The Cholesky decomposition of a symmetric matrix $C$ is defined by $C = U\,U^{\mathrm{T}}$, where $U$ is a triangular matrix. What is Chol($C$) in Eq. (4) ? Either one of those two triangular matrices, or what ?

6. Ll. 81-82. I presume the $\omega_{i,j}$'s are mutually independent ? Say it explicitly.

7. L. 109. Since no particular meaning would apparently be given to $F_i(x_j)$ for $i \neq j$, one index $i$ is sufficient (the double index may actually be confusing). I suggest to write $F_i(z)$, where $z$ is a dummy real argument.

8. End of caption of Fig.5. *dashed … lines → dotted …*

9. L. 241, values of variance of the observation error $\sigma^2$ do not make much sense without some appropriate scale of reference (for instance, the climatological variance of the solutions of the L96 model).

10. L. 386 and further below. Gaussian variable $G(x_1; -1, 2)$. The argument $x_1$ is here useless. And why not use the established notation $\mathcal{N}(-1, 4)$ for gaussian variables ($\mathcal{N}$(expectation, variance)) ?

11. L. 79, $I_N$ is ***the*** *... identity matrix*

12. L. 237, … *the model variable interpolated to location* … Which kind of interpolation ?

I cannot as Editor take a decision or even give a formal advice as long as the Interactive Discussion, to which any member of the scientific community can contribute, has not been closed. I nevertheless encourage the author, if he has not already done so, to start preparing a revised version of his paper, taking into account the comments and suggestions of the two referees, as well as my own. A revised version will have to come with a point-by-point response to all of these comments and suggestions. Should the author disagree with one particular comment, or decide not to follow one particular suggestion, he will have to state precisely his reasons for that.

---

## Author Response (AR1)

**Responses to Editor and Reviewers**

I would like to express my sincerest gratitude to the editor, Olivier Talagrand, for overseeing the review of my manuscript and for providing invaluable feedback. I would also like to thank both of my reviewers, Ian Groom and anonymous, for their thorough review, commentary, feedback and kind words. This review process has helped me substantially improve the quality and clarity of the manuscript. In recognition of the editor's and reviewers' efforts, I have included them in the Acknowledgement section of my revised manuscript.

**1  Responses to Reviewer 1 (Ian Groom)**

**1.1  General Comments**

**This paper presents a method for increasing the ensemble size during the assimilation step of ensemble DA. The method is interesting and the results show that it can improve performance. I have some suggestions for improving the clarity of the discussion though.**

Thank you, Ian, for your kind words, thorough review and valuable feedback. Your comments, particularly regarding Section 3, have greatly enhanced the clarity of my thoughts and the discussion. I have made every effort to address your comments and hope that my efforts will bring this manuscript closer to being accepted for publication in NPG.

**1.2 Major Comment**

**Section 3 could be improved. It attempts to show how PECE-GC could be valuable across a wide range of ensemble DA methods and gives 3 mechanisms whereby PECE-GC could improve performance. I found the whole section to be a bit too vague though. It might help to pick a few specific algorithms and show how PECE-GC could improve each one. For example, the RHF uses a piecewise-linear approximation of the likelihood function; PECE-GC improves this representation. As another example, many EnKFs assume joint Gaussianity of the state and observation vectors, and then approximate the means and cross-covariances of this distribution using an ensemble. If the prior is accurately represented then PECE-GC leads to improved estimates of these means and covariances.**

**Some things I found confusing about section 3: It starts with two-step ensemble DA, but not all EnKFs operate in a two-step manner. The three mechanisms seem to overlap; e.g. the likelihood function (3.1) is related to the observation operator (3.2), and to observation space ensemble statistics (3.3). Section 4 explains performance in light of these 3 mechanisms, so I also found it confusing.**

Thank you for pointing out this vagueness. As you have rightly pointed out, organizing the discussion around overlapping mechanisms instead of algorithms is confusing. I have consolidated the two overlapping mechanisms (the former 3.2 and 3.3) into a single mechanism (PESE-GC influences ensemble statistics). Furthermore, I have overhauled Section 3 so that there are algorithm-by-algorithm discussions. I have also replaced the confusing two-step discussion with a discussion based on Bayes' Rule. Section 4 has also been edited to only speak of 2 mechanisms instead of 3. The overhauled Section 3 is attached in the following pages.

[revised manuscript text omitted]

**1.3 Minor Comments**

**1.3.1 'Gaussian copula' is in the name of the method, but the concept of a Gaussian copula is not explained until section 5.**

Thanks for catching that. I have added the following paragraph in Section 2 (right after describing the 4-step PESE-GC procedure).

    "*Note that this four-stage assumes that the multivariate forecast distribution is Gaussian in probit space. This assumption arises from the use of a Gaussian resampling algorithm (the CAC2020 algorithm) to generate virtual probits. This assumption is equivalent to assuming that the multivariate forecast distribution has a Gaussian copula. As such, this four-stage procedure is called "Probit-space Ensemble Size Expansion for Gaussian Copulas".'*"

**1.3.2 Line 15/16: I struggle to see how 'small forecast ensembles result in limited representation of observation likelihood functions' is a distinct concept from sampling errors. PECE-GC seems to me to be a way to mitigate certain kinds of sampling errors.**

Thanks for catching this lack of clarity in the first lines of the introduction. Your interpretation is correct – PESE-GC mitigates certain kinds of sampling errors. I have updated the discussion in Section 3 to reflect that thinking (see attached pages). The updated introduction now reads:

    "*Geophysical forecast models are often computationally expensive to run. As a result, geophysical ensemble data assimilation (EnsDA) typically uses <100 forecast ensemble members. Such small forecast ensemble sizes result in sampling errors that degrade the performance of EnsDA. As such, low-cost methods that introduce additional ensemble members (henceforth, "virtual members") to the original forecast members (henceforth, "forecast members") have the potential to improve EnsDA.*"

**1.3.3 Lines 33/34: It might be more accurate to say that ensemble modulation preserves the first two moments of the ensemble, not that it assumes Gaussian statistics.**

You are correct. I have adjusted the paragraph in question. That paragraph now reads:

"*Ensemble modulation (Bishop and Hodyss, 2009, 2011; Bishop et al., 2017; Kotsuki and Bishop, 2022; Wang et al., 2021) is the fourth type of ensemble expansion method. Such methods expand ensembles by combining a localization matrix with the original ensembles' perturbations (see Bishop and Hodyss (2009) for details). Though the expanded ensemble possesses the same mean and variance as the original ensemble, the expanded ensemble's kurtosis can be much larger than the original ensemble's (see the Supplement). In other words, the expanded ensemble's kurtosis is likely biased. If nonlinear observation operators are applied on the expanded ensemble, this kurtosis bias will result in biased expanded ensemble observation statistics (personal communication with Craig Bishop and Lili Lei).*"

**1.3.4 Line 218: I think 0.05 model time units is usually interpreted as 6 hours, not 1 hour.**

Thanks for pointing this out. I noticed some inconsistencies in the literature regarding the interpretation of those time units. To prevent potential reader confusion, I have removed all conversions from L96 time units to hours. The relevant lines now read:

"*The L96 model uses 40 variables (i.e., 40 grid points in a ring), a forcing parameter value of 8 (i.e., F = 8), and a time-step of 0.05 L96 time units. The L96 time unit is henceforth referred to as τ. Forward time integration of the model is done via the Runge-Kutta fourth-order integration scheme. Every OSSE experiment runs for 5,500 cycles. Initial nature run states and the initial ensemble members are drawn from the L96's climatology.*
*Note that the 0.05τ time step has been arbitrarily taken to be either 1 hour (Anderson, 2019, 2023) or 6 hours (Lorenz, 2006; Kurosawa and Poterjoy, 2021). To avoid potential confusion, this paper will avoid converting L96*

Figures 7, 8 and 9 have also been modified to use $\tau$ instead of hours. I have also replaced all uses of hours with $\tau$.

**1.3.5 Line 259: The fifth-order GC function is rational, not polynomial.**

Thank you for catching this. The relevant line now reads:

"*The Gaspari-Cohn fifth-order rational function...*"

**1.3.6 Line 272: Missing citation for RTPS.**

Thank you for catching that. The citation is now added to the relevant line.

"*...or a relaxation-to-posterior-spread (RTPS; Whitaker and Hamill (2012))...*"

**1.3.7 I do not understand why PECE-GC has any impact on the performance of the stochastic EnKF with linear obs. With linear obs the stochastic EnKF update is entirely controlled by the ensemble covariance B, and PECE-GC does not change B (except in the PR configuration).**

Thank you for asking this question. This question helped me to understand why PESE-GC influenced the stochastic EnKF. PESE-GC specifically influences the performance of the serial stochastic EnKF. I have added the following paragraph to Section 3.1 to explain why.

"*Impact mechanism 1 also manifests for the serial stochastic EnKF. To see that, consider a situation with two observations, and recall that the serial stochastic EnKF uses random draws from a univariate Gaussian distribution to represent the likelihood function (one draw per ensemble member). For*

*small ensembles, only a few of those random draws are made. In other words, there are sampling errors in representing the likelihood function. The ensemble statistics resulting from assimilating the first observation are thus degraded by those sampling errors. This degradation then affects the assimilation of the second observation. The assimilation of more than two observations compounds such sampling issues. Since PESE-GC increases the ensemble size, more draws from the likelihood function are made, thus suppressing sampling errors. As such, in the absence of other factors, PESE-GC will improve the performance of the serial stochastic EnKF.*"

In contrast, I do not expect PESE-GC to have any impact on the stochastic EnKF that simultaneously assimilates all observations in one shot. I have added the following lines to Section 3.1 (around line 185) to explain why.

"*Also, the stochastic EnKF that assimilates all observations simultaneously (all-at-once stochastic EnKF) is immune to this effect – the chain of events described in the previous paragraph will not occur for the all-at-once stochastic EnKF.*"

**1.3.8   Line 402: 1,000,00 should probably be 1,000,000?**

Thank you for catching that typographical error. It has been fixed.

**1.3.9   The GC part of PECE-GC seems to only have been used in the PR configuration, is that right?  The EAKF, EnKF, and RHF all use PECE without the GC part?**

You are correct. To be clear, the EAKF, EnKF and RHF experiments effectively used the CAC2020 algorithm – this is a result of using PESE-GC with Gaussian marginals. It is only in the PR experiments that I am actually employing non-Gaussian marginals. To clarify matters, I have added the following lines to the paragraph starting in Line 256:

"*For each EnsDA algorithm, only 1 set of marginals are used with PESE-GC. When PESE-GC is used with the EAKF, EnKF or RHF algorithm, Gaussian*

*marginals are selected for all 40 model variables. PESE-GC with Gaussian marginals is identical to the CAC2020 algorithm. ==In other words, for the EAKF, EnKF and RHF experiments, the virtual ensemble members follow multivariate Gaussian distributions.== For the PR algorithm, the Gaussian-tailed rank histogram is selected as the marginal for every one of the 40 model variables. ==This means the PR experiments' virtual ensemble members follow multivariate non-Gaussian distributions.== Future work can investigate the impacts of using PESE-GC with Gaussian-tailed rank histograms (or kernel density estimates) with the EAKF, EnKF and RHF."*

**2 Responses to Reviewer 2**

**2.1 General Comments**

**This manuscript nicely proposed a method to expand the ensemble size in probit space. Unlike commonly used methods for ensemble size expansion, this PSES-GC ensemble size expansion method can generate virtual members from any multivariate forecast distribution with a Gaussian copula, by using the users' knowledge of the marginal distributions of forecast model variables. This PSES-GC method has been tested with tremendous scenarios, using the L96 model. Results show that PSES-GC can improve the performances of ensemble-based data assimilation methods when the actual ensemble size is small, or the marginal distributions improves the forecast model variable statistics, or the rank histogram filter is used with non-parametric priors. Expanding the ensemble size has advantages to ensemble-based DA in many aspects, while this PSES-GC is an interesting one. The manuscript is well written and pleasant to read. I have several specific comments as below.**

Thank you for your kind words and thorough review. Your comments have significantly improved this manuscript. I have made every effort to address your comments and hope that my efforts will bring this manuscript closer to being accepted for publication in NPG.

**2.2 Minor Comments**

**2.2.1 L28, I'd like to bring up two references here. Sun et al. (2022, 2024) use "analog" ensemble for paleoclimate data assimilation. The paleoclimate data assimilation could be a potential application for the ensemble size expansion, since very limited ensemble members are available.**
Sun, H., L. Lei, Z. Liu, L. Ning, and Z.-M. Tan, 2024: A hybrid gain analog offline EnKF for paleoclimate data assimilation. J. Adv. Model. Earth Syst., 16, e2022MS003414, doi: https://doi.org/10.1029/2022MS003414
Sun, H., L. Lei, Z. Liu, L. Ning, and Z.-M. Tan, 2022: An analog offline EnKF for paleoclimate data assimilation. J. Adv. Model. Earth Syst., 14, e2021MS002674, doi: https://doi.org/10.1029/2021MS002674

Thank you for your suggestions. I have added those references to the relevant paragraph in my introduction. That paragraph now reads:

"*A third type of ensemble expansion method is to search a historical catalog for forecast states similar to the current forecast or observations (Van den Dool, 1994; Tippett and Delsole, 2013; Monache et al., 2013; Wan and Van Der Merwe, 2000; Grooms, 2021; Sun et al., 2022). The virtual members resulting from this search have flow-dependent statistics. Though such methods are historically expensive to employ, ongoing research may render them affordable in the near future (e.g., Sun et al. (2024)).*"

**2.2.2 L33-34, "However, ensemble modulation assumes that forecast uncertainties have Gaussian statistics." This statement is not right clear to me. Could you explain a little bit why the modulation assumes Gaussian statistics? The modulation just applies localization, not really an ensemble size expansion.**

Thank you for pointing out this issue. Reviewer 1 (Ian Groom) also pointed out this issue. I agree with both of you that ensemble modulation does not assume Gaussian statistics.

I agree with you that modulation does apply localization. However, the resulting modulated ensemble is much larger than the original unlocalized ensemble. As such, I consider ensemble modulation as an ensemble expansion technique.

The relevant paragraph in the introduction has been edited to remove the mistaken assumption about Gaussian statistics (see below).

"*Ensemble modulation (Bishop and Hodyss, 2009, 2011; Bishop et al., 2017; Kotsuki and Bishop, 2022; Wang et al., 2021) is the fourth type of ensemble expansion method. Such methods expand ensembles by combining a localization matrix with the original ensembles' perturbations (see Bishop and Hodyss (2009) for details). ==Though the expanded ensemble possesses the same mean and variance as the original ensemble, the expanded ensemble's kurtosis can be much larger than the original ensemble's (see the Supplement). In other words, the expanded ensemble's kurtosis is likely biased. If nonlinear observation operators are applied on the expanded ensemble, this kurtosis bias will result in biased expanded ensemble observation statistics== (personal communication with Craig Bishop and Lili Lei).*"

**2.2.3 L89, it would be nice to have Eq. (6) of Chan et al. (2023) here. Then the reader can immediately see the difference between Eq. (6) of Chan et al. (2023) and the W in this manuscript.**

Thank you for your suggestion. I have added a simplified form of that equation. The relevant paragraph now reads:

"*Note that this study's (and the CAC2020's) **W** differs from that of Chan et al. (2023) (henceforth, "CAC2023"). This is because the ==CAC2023's **W** [defined in Eq. (6) of CAC2023]== generates virtual members with undesirable non-Gaussian properties. ==The CAC2023's **W** can be written as==*

$$\text{CAC2023's } \boldsymbol{W}: \quad W_{ij} \equiv \delta_{i,j} - \frac{1}{N_V} \tag{1}$$

*To illustrate the issue with the CAC2023's **W**, suppose 5 forecast members are drawn from a standard normal distribution...*"

**2.2.4 Section 2.2 and Figure 3, it is not straightforward to see how the cross-variable relationships are handled? Are the covariances of model state variables naturally conserved by performing PPI for each kind of state variable?**
Again at l132, does the adjustment conserve the covariance in model space?

Great questions! I am handling these questions together because they are related. If Gaussian marginals are chosen for PESE-GC, the PPIs and the adjustments to probit-space mean and covariance will not change the covariance in model space (i.e., the covariance is conserved). However, if non-Gaussian marginals are chosen for PESE-GC, the covariances in model space will likely be changed. The way the covariances change likely depends on what marginals are chosen and the distribution fitting procedure (e.g., moment-matching versus maximum likelihood). This subtlety may have interesting effects (e.g., may increase covariance matrix rank) and deserves future investigation.

I have modified the fifth paragraph of my concluding section to reflect the open question that you have posed (see below). Thank you for this comment!

"*Another general area for future work is to develop the PESE-GC algorithm further. First, given the importance of localization in practical EnsDA, future work can and should explore inserting localization into PESE-GC. Second, the validity of PESE-GC's Gaussian copula assumption can be assessed in the context of geophysical modelling and forecasting. If the Gaussian copula assumption is inappropriate, then non-parametric methods to generate virtual probits can be explored. Third, methods to detect the usage of misinformed parametric marginal distribution families deserve exploration. One possible detection method is to employ hypothesis testing on the marginal distributions. For example, if Gaussian distributions are selected for PESE-GC, then the Shapiro-Wilk test can be applied on the forecast ensemble to determine if the selection*

*is misinformed (e.g., Kurosawa and Poterjoy (2023)).* ==*Finally, the use of non-Gaussian marginals with PESE-GC may alter ensemble covariances between model variables. This possibility deserves future investigation.*=="

**2.2.5  L160-165, it is not quite clear the difference between the components A and C. Are they both about p(y|x)?**

Thank you for your question. Component A is about p(y|x), and component C is about the prior statistics of the simulated observations (i.e., related to p(x)).

Note that have I rewritten Section 3 in response to the concerns of Reviewer 1 (Ian Groom). Instead of three components, there are now two components – one for the observation likelihood, the other for the prior distribution. I have attached the re-written Section 3 in the following pages.

[revised manuscript text omitted]

**2.2.6 L239, any explanation for the choice of the error variance of 0.25 and 16 for the SQRT and SQUARE observations, given 1.0 for IDEN?**

Thank you for spotting this omission. Those values are taken from Anderson (2020) and I have updated the relevant sentence to treat that omission. The relevant sentence now reads:

> Following Anderson (2020), the $\sigma^2$ values for the IDEN, SQRT and SQUARE observations are set to 1.0, 0.25, and 16, respectively.

**2.2.7 L272, the reference for RTPS is missing.**

Thank you for catching that error. I have furnished the reference for RTPS (Whitaker and Hamill, 2012) in the manuscript.

**2.2.8 Section 4.4, here mechanisms 1-3 are used. It would be better to consistently use components A, B and C as in previous discussions.**

Thank you for your suggestion. I have replaced the three components with two mechanisms in response to Reviewer 1's comments. The discussion in Section 4 now uses the same terminology as in Section 3.

**2.2.9 L323-327, I don't quite understand here. I thought it should be more Gaussian if in probit space.**

Good question. The issue here is that Gaussian marginals are chosen for the PPIs used for PESE-GC in the EAKF, EnKF and RHF experiments. The Gaussian PPI transform can be shown to be a rescaling and recentering procedure. In other words, if the ensemble distribution in native model space is non-Gaussian, then the resulting distribution in probit space will also be non-Gaussian. In other words, the bad performance here is simply because inappropriate marginals are chosen for the PPI (the ensemble is probably following

a non-Gaussian distribution).

**2.2.10   L340-342, I am not convinced here. The RHF can be seen as a non-Gaussian filter.**

Thank you for your concern! I agree with you that the RHF is a non-Gaussian filter. However, a major difference between the RHF experiments and the PR experiments is in the the marginals chosen for PESE-GC – Gaussian for RHF and Gaussian-tailed rank histogram for PR. The issue here is PESE-GC with Gaussian marginals are used even though the true model-space forecast distribution is non-Gaussian. In other words, the expanded ensemble's statistics are worse than the forecast ensemble's statistics. PESE-GC with non-Gaussian marginals (e.g., Gaussian-tailed rank histogram marginals) is likely more appropriate in this situation. Even though the RHF is a non-Gaussian filter, if the ensemble statistics themselves are degraded, then the RHF's performance will suffer.

**2.2.11   It is nice to virtually increase the ensemble size. But it would be nicer to discuss the computational cost for the DA process, with increased ensemble sizes.**

Thank you for your suggestion. I have added the following paragraphs to Section 5.

"*It is also important to discuss the impacts of PESE-GC on the EnsDA process (i.e., the forecast step and analysis step). Since the virtual members are deleted before running forecast models (Figure 1), PESE-GC does not change the forecast step's computational cost. However, PESE-GC will increase the computational cost of the analysis step because 1) the number of observation operator calls scales linearly with the ensemble size, and 2) the EnsDA filter algorithm's (e.g., the EAKF algorithm) computational complexity scales with the ensemble size. Supposing the computational cost of the EnsDA filter algorithm scales linearly with ensemble size, then the computational cost of the EnsDA analysis step scales linearly with the number of ensemble members created by PESE-GC.*

*However, the increase in the computational cost associated with PESE-GC is likely far more affordable than running a larger forecast ensemble. This is because the computational cost of the forecast step often accounts for $\sim 80\%$ of the overall computational cost of the EnsDA process and the analysis step accounts for the remaining $\sim 20\%$. Consider the situation where PESE-GC quintuples the ensemble size. The overall computational cost of the EnsDA process will only increase by $\sim 80\%$. In contrast, if the forecast ensemble size is quintupled, then the overall computational cost of the EnsDA process increases by $\sim 400\%$. As such, PESE-GC is an alluring alternative to increasing the forecast ensemble size.*"

**3 Responses to the Editor**

**The two referees have evaluated positively the paper, and recommend acceptance, subject to minor revisions. Referee 1 (who has let his name known, and is Ian Grooms) has a major comment about Section 3 of the paper, which he finds a 'bit too vague', and for which he would like clearer explanations. He has in addition a number of minor, mostly editorial, comments. Referee 2 has also a number of minor comments, which have to do with both scientific and editorial aspects. I as Editor have also a number of comments (I mention that there seems to be a shift of a few units between the line numbers below and the ones mentioned by the referees).**

Thank you for summarizing the two reviewer's contributions and for your thorough review. I greatly appreciate the effort that you and the reviewers have spent on this manuscript. Your comments have significantly improved this manuscript and I have made every effort to address them. I hope that my efforts will bring this manuscript closer to being accepted for publication in NPG.

**3.1 Conceptual comments**

**3.1.1 I wonder in particular what the virtues of the extended PESE-GC ensemble are. I am sure these virtues are described in the previous publications of the author (in particular Chan et al., 2020, and Chan, 2022), but it could be useful to say a little more in the present paper for readers who are new to that method. ... In particular does the CAC2020 algorithm (subsection 2.1) introduce additional properties of interest ? Additional explanations on those points, even succinct, could be useful (this aspect may have to do with Referee 1's major comment).**

Thank you for pointing out this deficiency. My previous publications do not discuss PESE-GC at all. Instead, my previous publications discuss an efficient and embarrassingly parallel Gaussian resampling scheme. The Gaussian nature of

that scheme is the scheme's primary weakness. This manuscript extends that Gaussian resampling scheme to handle non-Gaussian distributions. I have re-purposed Section 2.1.5 to describe the properties of that Gaussian resampling scheme (the CAC2020 algorithm). The following paragraphs have been added to Section 2.1.5 as part of the repurposing.

"*The CAC2020 algorithm also produces expanded ensembles with same ensemble means and ensemble covariances as the forecast ensembles. In other words, the rank of the expanded ensemble's covariance matrix is the same as that of the forecast ensemble. Future work can explore ways to incorporate localization into the expanded ensemble's covariance matrix.*

*Furthermore, the CAC2020 algorithm always generates Gaussian-distributed virtual members: even if the actual forecast distribution is highly non-Gaussian, the virtual members' distribution will still be Gaussian. The CAC2020 algorithm thus degrades the ensemble statistics in situations where the forecast distribution is non-Gaussian. This degradation limits the usefulness of the CAC2020 algorithm for situations with non-Gaussian forecast distributions.*

*Note that, except for the mean and covariance, the expanded ensemble's central moments (i.e., higher-order moments; e.g., skewness) likely differ from the forecast ensemble's. More specifically, the expanded ensemble's central moments will be closer to those of Gaussian distributions (e.g., zero skewness) than the forecast ensemble's central moments. This is because the virtual members are effectively drawn from a Gaussian distribution. If the forecast distribution is indeed a Gaussian distribution, then the expanded ensemble likely has better moments than the forecast ensemble.*"

**3.1.2 From what I understand, specific properties of the extended PESE-GC ensemble are that it has the same mean and covariance matrix as the forecast ensemble, that it preserves through PPI the marginal distributions of the forecast model variables, and that it is numerically very economical.**

Thank you for your feedback. I would like to highlight that PESE-GC specifically conserves the mean and covariance in probit space (i.e., the transformed space). If the marginals chosen are Gaussian, then PESE-GC does conserve

the mean and covariance in the original space (i.e., untransformed). However, if non-Gaussian marginals are chosen, then it is not immediately clear what PESE-GC does to the mean and covariance in the original space. I have added a sentence to the fifth paragraph of my concluding section to point out this area of future work (see below).

"*Another general area for future work is to develop the PESE-GC algorithm... Finally, the use of non-Gaussian marginals with PESE-GC may alter ensemble covariances between model variables. This possibility deserves future investigation.*"

**3.1.3 Would any other ensemble expansion method that had the same properties (maybe there is not any) be as useful?**

I am not aware of any other ensemble expansion methods with the numerical economy and generality of the PESE-GC algorithm. To highlight this, I have added a sentence to the end of the sixth paragraph in Section 1 (see below).

"*The shortcomings of existing ensemble expansion methods motivate... To the author's knowledge, no other ensemble expansion method simultaneously possesses the same efficiency, scalability, generality and flow-dependency as PESE-GC.*"

**3.1.4 My understanding is that the CAC2020 algorithm is implemented between steps 3 and 4 of the PESE-GC procedure (ll. 122-126). Say it there. And, if I am mistaken, additional appropriate information will be useful.**

Thank you for pointing out my lack of clarity. I have added two sentences stating the usage of the CAC2020 algorithm in stage 3 of the PESE-GC procedure. The first sentence occurs right after the in-text description of the 4 stages of PESE-GC:

"*To be clear, the CAC2020 algorithm is implemented and used in stage 3.*"

The second sentence occurs inside the caption of Figure 3:

"*...The details of these stages are described in Section 2.2.* *Note that the CAC2020 algorithm is applied in stage 3 (i.e., between panels c and d)**.*"

**3.1.5 Step 3 of PESE-GC and subsection 2.3. Is the need to adjust the mean and variance of each variable to 0 and 1 due only to the finiteness of the forecast ensemble ? If yes, say it explicitly. If no, explain more clearly.**

Thank you for your suggestion. Your understanding is correct. I have added a sentence saying that the finiteness of the forecast ensemble is the source of the problem. That sentence is in the first paragraph of Section 2.3.

"*PESE-GC requires forecast ensemble probits with zero mean and unity variance. Otherwise, the resulting virtual members will disobey the marginal distributions fitted in PESE-GC's step 1.* *However, because the forecast ensemble size is finite, the forecast ensemble's probits may have non-zero mean and non-unity variance.* *To illustrate...*"

**3.2 Editorial comments**

**3.2.1 Eq. (2), denominator on the rhs $N_E \rightarrow N_V$**

Thank you for catching my typographical error. I have fixed that error.

**3.2.2 Eq. (4). What does Chol(.) exactly mean here ? The Cholesky decomposition of a symmetric matrix C is defined by C = U UT, where U is a triangular matrix. What is Chol(C) in Eq. (4) ? Either one of those two triangular matrices, or what ?**

Good question. I am specifically using the lower-triangular matrix is produced by the Cholesky decomposition (i.e., U). I have corrected my description of the Chol($*$) operator:

"... $\text{Chol}(\cdot)$ represents Cholesky decomposition (which produces a lower-tri-angular matrix), ..."

**3.2.3 Ll. 81-82. I presume the $\omega_{i,j}$ are mutually independent ? Say it explicitly.**

You are correct. I have added a statement of that mutual independence after Eq. (7):

"where all $\omega_{i,j}$s are identically and independently distributed (i.i.d.) samples drawn from the standard normal distribution. In other words, all $\omega_{i,j}$s are mutually independent."

**3.2.4 L. 109. Since no particular meaning would apparently be given to $F_i(x_j)$ for $i \neq j$, one index $i$ is sufficient (the double index may actually be confusing). I suggest to write $F_i(z)$, where $z$ is a dummy real argument.**

Thank you for your suggestion. However, there is no index $j$ in the equation you've referenced – there are two occurences of the same index $i$. I deliberately put two $i$ indices on $F_i(x_i)$ to highlight that the CDF varies with the model variable in question ($i$ is the model variable index). I have added the following sentence to the paragraph before Eq. (10):

"...and $\phi_i$ represents the $i$-th model probit. The double appearance of index $i$ in $F_i(x_i)$ is deliberate – the CDF varies with the chosen model variable. Note that..."

**3.2.5 End of caption of Fig.5. *dashed ... lines* → *dotted ...**

The correction has been made. The last sentence in Fig. 5's caption now reads

"The true CDFs and x-y relationships are plotted in panels b and c with dotted black lines."

**3.2.6 L. 241, values of variance of the observation error $\sigma^2$ do not make much sense without some appropriate scale of reference (for instance, the climatological variance of the solutions of the L96 model).**

Thank you for pointing this out. Reviewer 2 also made a similar comment. Those $\sigma^2$ values are lifted from Anderson (2020). The relevant sentence now reads:

> *Following Anderson (2020), the $\sigma^2$ values for the IDEN, SQRT and SQUARE observations are set* to 1.0, 0.25, and 16, respectively.

**3.2.7 L. 386 and further below. Gaussian variable $G(x_1; -1, 2)$. The argument $x_1$ is here useless. And why not use the established notation N(-1, 4) for gaussian variables (N(expectation, variance)) ?**

Thank you for pointing this out. However, my use of $G(x_1; -1, 2)$ instead of $N(-1, 2)$ is deliberate: $N(-1, 2)$ represents the Gaussian distribution, but $G(x_1; -1, 2)$ represents the **PDF** of the Gaussian distribution. To illustrate, consider the equation

$$p(x) = N(-1, 2).$$

When (inexperienced) readers look at that equation, they may get confused and ask "where did the $x$ go in the rhs?".

If I am saying "$x$ is drawn from $N(-1, 2)$", then the notation is (to my understanding) $x \sim N(-1, 2)$. However, because the equation is describing the PDF of $x_1$, not sampling $x_1$, then $G(x_1; -1, 2)$ is more appropriate. As such, I would like to keep my notation as is.

**3.2.8 L. 79, $I_N$ is the . . . identity matrix**

I have made the correction (an→the). The sentence now reads:

"...$\boldsymbol{I_{N_E}}$ is the $N_E \times N_E$ identity matrix"

**3.2.9  L. 237, ...  the model variable interpolated to location ... Which kind of interpolation ?**

Thank you for catching that. Linear interpolation is used. The sentence now reads:

[revised manuscript text omitted]

---

## Editor Decision (ED1)

Two referees have sent their evaluations. They are the same as the referees of the previous version (referee 1 is I. Grooms, and referee 2, who does not want to remain anonymous any more, is Lili Lei, from Nanjing University).

Both of them consider the author has satisfactorily responded to their concerns, and recommend acceptance of the paper. Referee 1 just mentions that there are typos to be corrected (the paper will in any case go through copy-editing).

I follow the referees' recommendation, and accept the paper. I however as editor still have a few suggestions for modifications.

1. L. 268. It would be preferable to give a scale of comparison for the variances of the observation errors. My colleague Mohamed Jardak and myself (Jardak and Talagrand, 2018) found a 'climatological' variance of $10^3$ for the Lorenz 96 model, to be compared with the value 1 used here by the author for the IDEN observations (but the numerical conditions of the experiments may not be the same).

2. L. 217. *Figure 5 indicates that the virtual members have better ensemble statistics than the forecast ensemble.* What do you mean by *better* ? The CDFs shown in Fig. 5 from the virtual members are smoother than the CDFs obtained from the forecast members. Is that what you mean, or what ?

3. Ll. 73-74. I suggest to state more precisely what the notation Chol($C$) exactly means. I understand it denotes the lower triangular matrix of the Cholesky decomposition $C = U\,U^{\mathrm{T}}$ of the matrix $C$, but is it $U$ or $U^{\mathrm{T}}$ that is lower triangular (that may be irrelevant, but may nevertheless matter for a reader who wants to implement the algorithm) ?

REFERENCE

Jardak, M., and O. Talagrand, Ensemble variational assimilation as a probabilistic estimator – Part 1: The linear and weak non-linear case, 2018, *Nonlin. Processes Geophys.*, **25**, 565-587, https://doi.org/10.5194/npg-25-565-2018.

---

## Author Response (AR2)

**Responses to Editor and Reviewers**

Thank you, Editor (Dr. Olivier Talagrand), for overseeing the review of my manuscript, for providing invaluable feedback, and for granting my request for a deadline extension. I am also indebted to my Reviewers (Dr. Ian Grooms and Dr. Lili Lei) for their thorough review, commentary, feedback and kind words. This review process has substantially and obviously improved my manuscript's quality and clarity. I have expressed my gratitude to Dr. Talagrand, Dr. Grooms, and Dr. Lei in my acknowledgements.

**1 Responses to Reviewer 1 (Ian Grooms)**

Thank you, Ian, for taking the time to re-review my manuscript thoroughly and for sending me an email regarding my typographical errors. Because I could not locate your comments on EGUSphere, I am responding to your emailed comments below. I have made every effort to address your comments in the hopes that this will bring my manuscript to acceptance.

**1.1 Minor Comments**

**1.1.1 page 8: "Note that this four-stage assumes" -> "Note that this four-stage procedure assumes"**

I have corrected this error. Thank you for catching that.

**1.2 Line 301ff: The Lorenz & Emmanuel 1998 paper on the L96 model identifies 0.05 model time units as comparable to 6 hours of synoptic-scale weather not arbitrarily but based on matching the predictability characteristics of the two systems. I searched the Anderson 2019 and 2023 papers and didn't see where Jeff says that 0.05 MTU equals 1 hour. This is not important, but I wanted to give you the feedback.**

Thank you for the feedback. I have removed the offending paragraph and modified the previous paragraph slightly (see below).

"*The L96 model uses 40 variables (i.e., 40 grid points in a ring), a forcing parameter value of 8 (i.e., F = 8), and a time-step of 0.05 L96 time units. The L96 time unit is henceforth referred to as $\tau$.* All results in this paper will be displayed and discussed in terms of $\tau$. *Forward time integration of the model is done via the Runge-Kutta fourth-order integration scheme. Every OSSE experiment runs for 5,500 cycles. Initial nature run states and the initial ensemble members are drawn from the L96's climatology.* "

**1.3 Thanks for the acknowledgment, but you acknowledge Groom not Grooms! Just a typo.**

Thanks you for spotting that! The issue has been fixed.

**2  Responses to Reviewer 2 (Lili Lei)**

Thank you, Lili, for re-reviewing my manuscript, and for recommending acceptance. I am grateful for your thorough review and suggestions. I have acknowledged your contributions to this manuscript in my Acknowledgements section.

**3   Responses to the Editor (Lili Lei)**

**3.1   General comments**

**Two referees have sent their evaluations. They are the same as the referees of the previous version (referee 1 is I. Grooms, and referee 2, who does not want to remain anonymous any more, is Lili Lei, from Nanjing University).**
**Both of them consider the author has satisfactorily responded to their concerns, and recommend acceptance of the paper. Referee 1 just mentions that there are typos to be corrected (the paper will in any case go through copy-editing).**
**I follow the referees' recommendation, and accept the paper. I however as editor still have a few suggestions for modifications.**

Thank you for summarizing the two reviewers' comments, for your thorough review, and for overseeing the review of my manuscript. I have added an acknowledgement of Lili in my manuscript's Acknowledgements section. I have made every effort to incorporate your suggestions in the hopes of my manuscript's acceptance.

**3.2   Editorial Suggestions**

**3.2.1   L. 268. It would be preferable to give a scale of comparison for the variances of the observation errors. My colleague Mohamed Jardak and myself (Jardak and Talagrand, 2018) found a 'climatological' variance of $10^3$ for the Lorenz 96 model, to be compared with the value 1 used here by the author for the IDEN observations (but the numerical conditions of the experiments may not be the same).**

Thank you for encouraging me to explore the climatological variances of my three observation operators. I have estimated those variances by constructing a 10,000-member climatological ensemble. This ensemble is constructed by initializing those members with noise samples drawn from $N(\mathbf{0}, \mathbf{I})$, and then integrating for 5000 time steps (i.e., 100 model time units).

[Figure]

**Figure 1:** Ensemble-estimated climatological prior variances of (a) the IDEN observation operator, (b) the SQRT observation operator, and (c) the SQUARE observation operator. Because the observation sites are in-between Lorenz 1996 model grid points, the climatological variances vary depending on where the sites are between the two points. As such, the variances here are plotted as a function of the interpolation weight from the grid point to the right of the observation site.

Note that my variances are complicated by the fact that I am linearly interpolating from the 40 Lorenz model grid points to in-between grid point values. As such, I have computed the climatological variances as a function of the interpolation weight applied to the grid point to the right of the observation site

(Figure 1).

The IDEN observation's climatological variance is between 6.6 to 13.5. This means my observation error variance for the IDEN observations is between 7.4% to 15% of the climatological variance. As for the SQRT observations, my observation error variance of 0.25 is between 10% to 17% of the SQRT observation's climatological variance. Finally, the observation error variance for my SQUARE observations is 16, which is between 2% to 6% of the corresponding climatological error variance.

I have added the following sentence to the paragraph describing the observation error variance (Section 4.1):
"... the chosen $\sigma^2$ for the IDEN, SQRT and SQUARE observations are 1.0, 0.25, and 16, respectively. ==Note that IDEN's $\sigma^2$ is between 7.4% to 15% of the climatological IDEN error variance, SQRT's $\sigma^2$ is between 10% to 17% of the climatological SQRT error variance, and SQUARE's $\sigma^2$ is between 2% to 6% of the climatological SQUARE error variance.=="

**3.2.2  L. 217. Figure 5 indicates that the virtual members have better ensemble statistics than the forecast ensemble. What do you mean by better ? The CDFs shown in Fig. 5 from the virtual members are smoother than the CDFs obtained from the forecast members. Is that what you mean, or what ?**

Thank you for encouraging me to be clearer on the matter. I have clarified my meaning in my manuscript (see below).
"... Figure 5 indicates that ==the virtual members' CDFs and x-y relationship are closer to the true CDFs and relationship than those of the forecast members – the virtual members' curves have visibly less distances from the true curves than the forecast members' curves. In other words,== the virtual members have better ensemble statistics than the forecast ensemble. This improvement in ensemble statistics ..."

**3.2.3 Ll. 73-74. I suggest to state more precisely what the notation Chol(C) exactly means. I understand it denotes the lower triangular matrix of the Cholesky decomposition C = U UT of the matrix C, but is it U or UT that is lower triangular (that may be irrelevant, but may nevertheless matter for a reader who wants to implement the algorithm)?**

Thank you for encouraging me to be clearer on the subject. I have clarified that in my manuscript (see below).

*"The CAC2020 algorithm constructs $N_V$ virtual members from $N_E$ ensemble members using a three-step procedure. First, an $N_E \times N_V$ matrix of linear combination coefficients ($\mathbf{E}$) is generated by evaluating*

$$\mathbf{E} \equiv \gamma \mathbf{1}_{N_E \times N_V} + \mathbf{L}_{C_E} \left\{ \mathbf{L}_{WW^\top} \right\}^{-1} \mathbf{W}. \tag{1}$$

*Here,*

$$\gamma \equiv \frac{1}{N_V} \left( \sqrt{\frac{N_E + N_V - 1}{N_E - 1}} - 1 \right), \tag{2}$$

*$\mathbf{1}_{N_E \times N_V}$ is an $N_E \times N_V$ matrix of ones, $\mathbf{L}_{C_E}$ is a lower-triangular matrix obtained from the Cholesky decomposition of $\mathbf{C_E}$ (note that $\mathbf{C_E} = \mathbf{L}_{C_E} \left( \mathbf{L}_{C_E} \right)^\top$), $\mathbf{C_E}$ is an $N_E \times N_E$ matrix defined by*

$$\mathbf{C_E} \equiv \frac{N_V}{N_E - 1} \mathbf{I}_{N_E} - \gamma^2 N_V \mathbf{1}_{N_E \times N_E}, \tag{3}$$

*$\mathbf{I}_{N_E}$ is the $N_E \times N_E$ identity matrix, $\mathbf{1}_{N_E \times N_E}$ is an $N_E \times N_E$ matrix where every element is one, $\mathbf{L}_{WW^\top}$ is a lower-triangular matrix obtained from the Cholesky decomposition of $\mathbf{WW}^\top$ (note that $\mathbf{WW}^\top = \mathbf{L}_{WW^\top} \left( \mathbf{L}_{WW^\top} \right)^\top$), and $\mathbf{W}$ is an $N_E \times N_V$ matrix whose $(i, j)$-th element is defined by...."*